# A multi-modal MRI study of the central response to inflammation in rheumatoid arthritis

Andrew Schrepf[1], Chelsea M. Kaplan[1], Eric Ichesco[1], Tony Larkin[1], Steven E. Harte[1], Richard E. Harris[1], Alison D. Murray [2], Gordon D. Waiter[2], Daniel J. Clauw[1] & Neil Basu[2,3]

It is unknown how chronic inflammation impacts the brain. Here, we examined whether higher levels of peripheral inflammation were associated with brain connectivity and structure in 54 rheumatoid arthritis patients using functional and structural MRI. We show that higher levels of inflammation are associated with more positive connections between the inferior parietal lobule (IPL), medial prefrontal cortex, and multiple brain networks, as well as reduced IPL grey matter, and that these patterns of connectivity predicted fatigue, pain and cognitive dysfunction. At a second scan 6 months later, some of the same patterns of connectivity were again associated with higher peripheral inflammation. A graph theoretical analysis of whole-brain functional connectivity revealed a pattern of connections spanning 49 regions, including the IPL and medial frontal cortex, that are associated with peripheral inflammation. These regions may play a critical role in transducing peripheral inflammatory signals to the central changes seen in rheumatoid arthritis.

[1] Chronic Pain and Fatigue Research Center, Department of Anesthesiology, University of Michigan Health System, Domino's Farms, Lobby M, PO Box 385, 24 Frank Lloyd Wright Drive, Ann Arbor, MI 48106, USA. [2] School of Medicine, Medical Sciences and Nutrition, Foresterhill, University of Aberdeen, Aberdeen AB25 2ZD, UK. [3] Institute of Infection, Immunity & Inflammation, University of Glasgow, Glasgow G12 8QQ, UK. Correspondence and requests for materials should be addressed to A.S. (email: aschrepf@med.umich.edu)

When people experience injury or infection, their behaviour and perceptions change dramatically. These changes tend to be conserved across mammalian species and include increased sensitivity to pain, fatigue and cognitive dysfunction. These are collectively referred to as 'sickness behaviours'[1]. The primary mechanisms through which these changes occur are inflammatory—molecular messengers of damage and infection signal the central nervous system (CNS) through afferent nerves, sentinel cells in the circumventricular organs and transport across the blood–brain barrier (BBB)[1]. However, the neural correlates of inflammation in chronic conditions have scarcely been explored. Chronic conditions characterized by both heightened inflammation and symptoms like fatigue and neurocognitive deficits include diabetes, cancer, dementia and rheumatoid arthritis (RA)[2–6]. Experimental paradigms in small samples of healthy individuals who receive immunogenic substances have demonstrated transient alterations in functional connectivity of the brain[7,8]. These effects, however, represent the response of the resilient CNS to acute inflammatory insults, and studies of how symptoms change over longer-term administration of immunogenic substances suggest potentially important differences in the central response to chronic vs. acute inflammation[9]. Characterizing the neuro–immune axis in chronic inflammatory conditions represents a crucial link in determining how repeated inflammatory insults result in debilitating symptoms in RA.

In this study, we leverage the natural fluctuations in levels of peripheral inflammation that are a feature of RA in order to determine how inflammation affects brain function and structure. RA is ideal for these purposes because the condition is archetypically inflammatory and chronic[6], with most sufferers experiencing periodic disease flares wherein both circulating inflammatory markers and symptoms become elevated. Our primary objectives were to identify specific brain regions that become more or less connected with each other as levels of inflammation increase. This was accomplished by first conducting a whole-brain search using Independent Component Analyses (ICA) to identify relationships with established neural networks. Second, we expanded on this approach using graph theoretical analyses to identify relationships with inflammation across the entire functional connectome. Finally, we employed voxel-based

morphometry (VBM) to identify areas where grey matter (GM) volume is increased or decreased in conjunction with levels of inflammation. To confirm the validity of our results, we repeated ICA analyses in the same individuals 6 months later. Our purpose was to identify, among patients with RA, brain regions and patterns of connectivity that are important in the central response to peripheral inflammation. We find that the left inferior parietal lobule (IPL) and medial prefrontal cortex (mPFC) play critical roles in the altered patterns of brain connectivity associated with peripheral inflammation in RA.

## Results

**Sample.** Fifty four RA patients underwent magnetic resonance imaging (MRI) at baseline and at 6 months (mean age 54.94 ± 11.41 years; $n = 41$ (76%) female; mean disease duration 11.49 ± 8.64 years) of a total of 335 patients who were approached (Supplementary Figure 1). There was variation in the primary clinical indication of inflammation, erythrocyte sedimentation rate (ESR; range: 2–62 mm/h at the baseline visit) and clinical presentation of RA. Clinical characteristics, including ESR, C-reactive protein (CRP), Disease Activity Score −28 (a composite of swollen joint counts, ESR and self-rated health)[10], and symptom levels at each time are displayed in Fig. 1 (and see Supplementary Table 1). A number of anti-inflammatory medications were used by the sample (Supplementary Table 2).

**Seed network to whole-brain connectivity analysis.** Eight functional brain networks identified through ICA[11] were used as seeds in a whole-brain search to identify specific brain regions that become more positively or negatively connected to the seed network at higher levels of peripheral inflammation (ESR). These analyses revealed several significant associations between higher ESR and positive functional brain connections and a single association with a negative functional connection. Four of the eight networks (default mode network (DMN), dorsal attention network (DAN), salience network (SLN), medial visual network (MVN)) showed these relationships (Table 1 and Fig. 2). Two brain regions showed connections to multiple networks, roughly comprising the left (L) IPL /L angular gyrus and the bilateral mPFC (Fig. 3). The other four networks (sensorimotor network (SMN), lateral visual network, left and right fronto-parietal

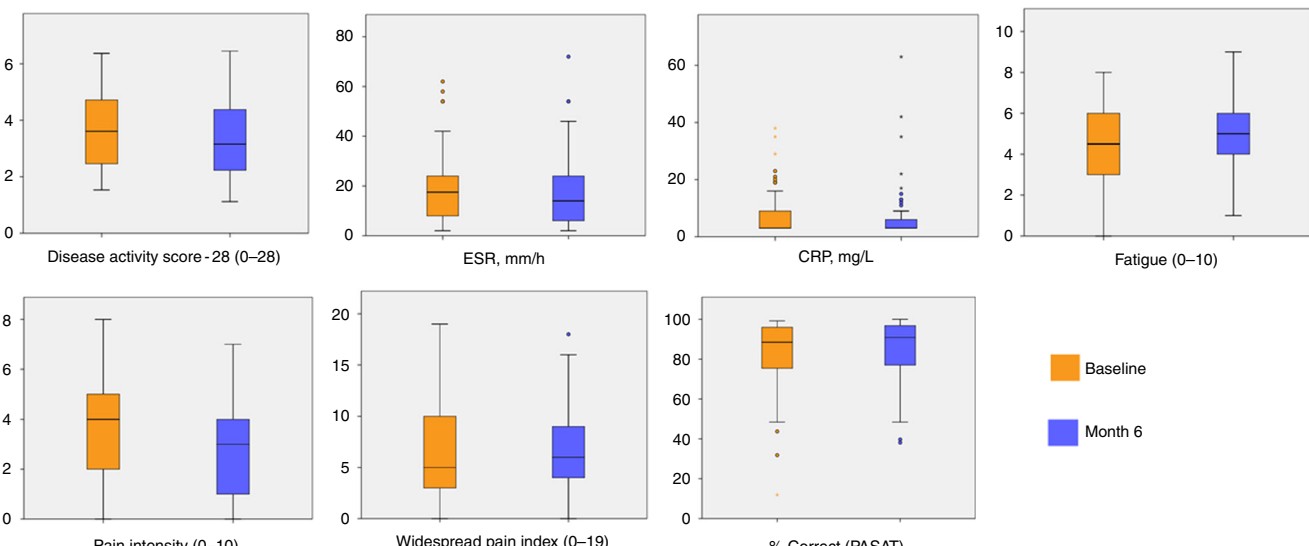

**Fig. 1** Box plots of clinical characteristics and symptoms at baseline and 6 months for 54 rheumatoid arthritis patients. The center line represents the median value, the lower bound of the box represents the 25th percentile, the upper bound of the box the 75th percentile, and the whiskers represent minimum and maximum values, or 1.5 times the interquartile range in the presence of outliers, which are indicated by dots or asterisks

**Table 1 Seed network to whole-brain connectivity analyses at baseline**

| Seed network region (direction of association) | MNI coordinates (x, y, z) | Z score | Cluster size (voxels) | p-value, FDR |
|---|---|---|---|---|
| Default mode network | | | | |
| L IPL (+) | −36, −76, 54 | 4.34 | 117 | 0.049[a] |
| R mid/sup frontal gyrus (+) | 38, 52, −2 | 4.00 | 135 | 0.049 |
| R posterior cingulate (−) | 16, −52, 30 | 4.28 | 172 | 0.008[a] |
| Dorsal attention network | | | | |
| L IPL (+) | −36, −76, 54 | 5.10 | 675 | <0.001[a] |
| R mPFC (+) | 12, 62, −12 | 4.50 | 245 | 0.001[a] |
| Medial visual network | | | | |
| L IPL (+) | −48, −56, 56 | 4.07 | 240 | 0.001[a] |
| Salience network | | | | |
| L IPL/angular gyrus (+) | −52, −58, 24 | 4.53 | 240 | 0.002[a] |
| L mPFC (+) | −2, 54, −16 | 4.02 | 171 | 0.009 |

Cluster-corrected false discovery rate (FDR) p-values are displayed. Analyses control for age and sex
[a]Also associated with higher levels of C-reactive protein (CRP); Spearman's rank procedure, $p < 0.05$

perception/language networks) showed no significant relationships. As noted in Table 1, six of the eight identified patterns of connections between regions and seed networks were also associated with higher levels of CRP. Three of these relationships were replicated at the second session with ESR at a trend level (L IPL to DMN, L IPL to DAN, right (R) mPFC to DAN) while five of the eight identified connections were replicated with CRP (L IPL to DMN, R PCC to DMN, L IPL to DAN, R mPFC to DAN, L IPL to MVN). See Fig. 4 and Table 2.

**Network-Based Statistic**. To explore more detailed patterns of inflammation-linked functional connectivity, all pairwise connections between 264 brain regions[12] were examined for subsets of brain regions (nodes) that are positively connected to one another (edges) at higher levels of peripheral inflammation. This approach uses a Network-Based Statistic (NBS)[13] derived from graph theoretical network analysis[14] to identify the relevant nodes and edges. Higher levels of ESR were associated with a significant NBS configuration comprised of 49 nodes and 54 edges. These are depicted visually in Fig. 5, and the nodes are listed in Table 3 (edges in Supplementary Table 3). Using the consensus network assignment of Power's atlas[12], these nodes belonged most frequently to the fronto-parietal task control (nine nodes), default mode (eight nodes) and salience (seven nodes) networks.

The L IPL/angular was well-represented in the NBS configuration, as five of the eight nodes that correspond approximately to the L IPL/angular region in the Power's atlas were present. Three of these five nodes were located in the L IPL region identified through the network to whole-brain connectivity analyses (regions 86, 87, 177). Similarly, three nodes were located in the medial frontal/orbital region, one of which was located in the medial frontal region identified through the network to whole-brain connectivity analyses (region 75). The three nodes that overlapped with the ICA findings in the IPL collectively participated in 10 of the 54 edges in the 'inflammation configuration', while the medial frontal node participated in three. Edges showed a diverse range of connections, including connections within the L IPL/L angular and connections from the L IPL to regions of the putative pain matrix such as the dorsal anterior cingulate cortex and anterior insula.

**Hub analyses**. Brain regions can be defined as hubs if they a large number of rich connections to other brain regions, a measure called eigenvector centrality (EC)[15]. Four nodes located in the L IPL region identified through seed network to whole-brain analyses were classified as hubs. Of the 49 nodes in the NBS inflammation configuration, 17 were classified as hubs. These are noted in Table 3 (all identified hubs are shown in Supplementary Table 4).

**EC and the inflammation configuration**. The EC score, a measure of overall hub-like qualities, derived from the 49 NBS nodes was positively associated with ESR ($r = 0.466$, $p < 0.001$; Pearson correlation). This indicates that the nodes comprising the 'inflammation configuration' are increasingly hub-like in their role in the whole-brain functional connectome in individuals with higher levels of peripheral inflammation.

**Region-of-interest VBM analyses**. Of the significant functional connectivity regions identified in Table 1, higher levels of ESR were associated with reduced GM volumes of the L IPL ($B = -0.00032$; multiple linear regression; $p = 0.038$ age/sex/total intracranial volume [TIV] corrected and posterior cingulate cortex ($B = -0.00050$; multiple linear regression; $p = 0.024$). This result did not meet false discovery rate (FDR) correction, a method for controlling for multiple comparisons ($p > 0.05$ FDR). Results of each region of interest (ROI) analysis are shown in Table 4.

**Whole-brain VBM**. One region of reduced GM volume was identified that was associated with higher levels of ESR. This was a region that included the L Cerebellum and L lingual gyrus (MNI coordinates: −4, −85, −21; z value: 4.07; cluster size: 2213 voxels; FDR-corrected $p < 0.001$). There were no relationships with increased GM volume.

The association between ESR and the GM volume of the L IPL was replicated in session 2 6 months later ($B = -0.00031$; multiple linear regression; $p = 0.035$ age/sex/TIV corrected), whereas the relationship with the posterior cingulate cortex was not ($B = -0.00029$; multiple linear regression; $p = 0.13$). Similar to the baseline results, this finding in the L IPL did not meet FDR correction ($p > 0.05$ FDR).

**Associations with symptoms**. Different clinical domains of RA were also associated with identified functional connections. Increased functional connectivity between the DAN and R mPFC correlated with increased levels of fatigue ($r = 0.46$), pain intensity ($r = 0.36$) and diffuse pain ($r = 0.34$; all $p < 0.05$; Pearson correlation). Enhanced functional connectivity of the SLN–L IPL/angular ($r = 0.34$), and DMN–L IPL ($r = 0.28$) were associated with more diffuse pain (both $p < 0.05$; Pearson correlation). Increased SLN–L mPFC connectivity was associated with more diffuse pain ($r = 0.27$) and a lower percentage of correct responses on the paced auditory serial addition test (PASAT; $r =$

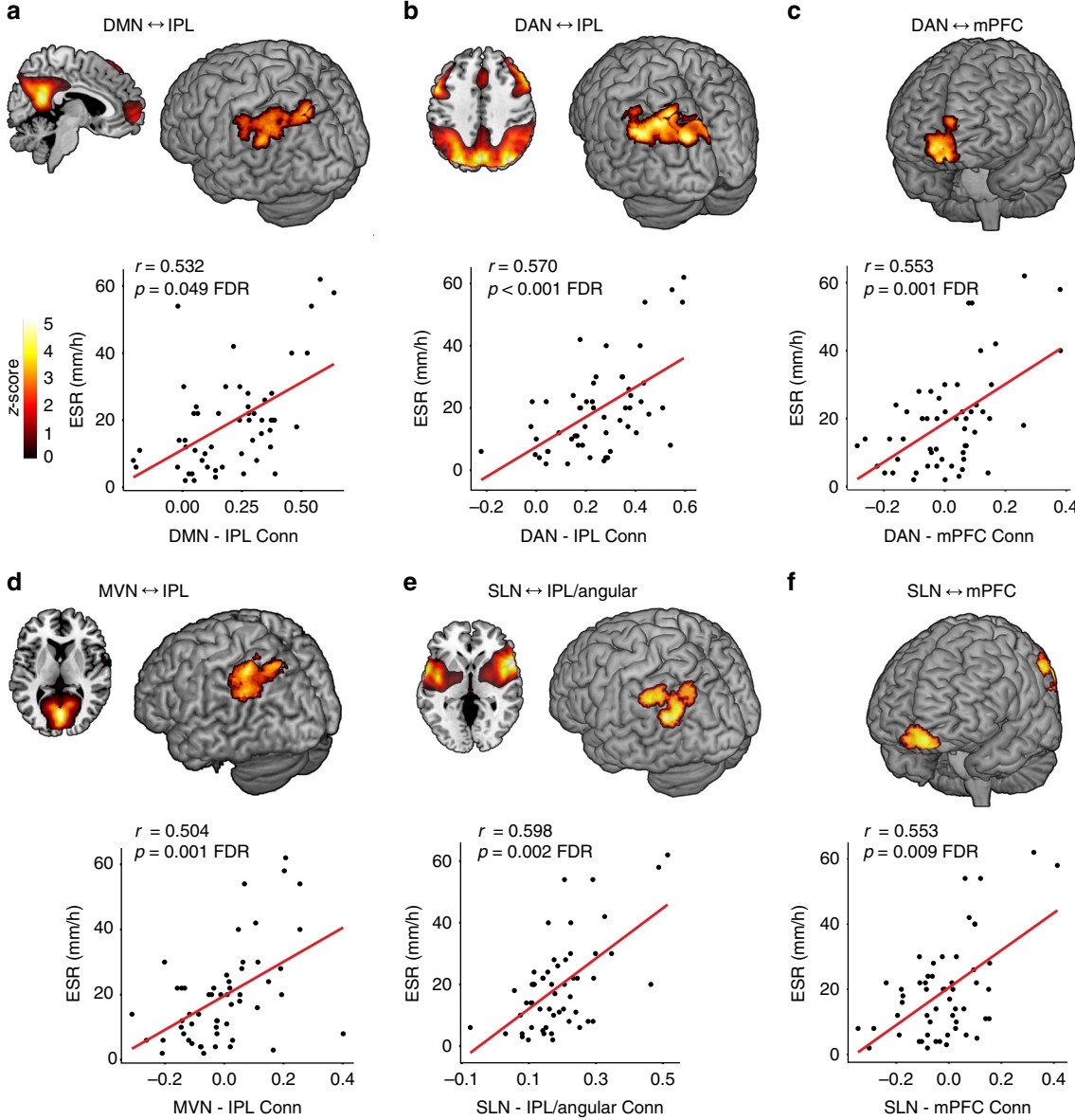

**Fig. 2** Inflammation-associated functional connectivity patterns. Brain regions (medial prefrontal cortex [mPFC], inferior parietal lobule [IPL]) showing positive connections to the default mode network (DMN; **a**), dorsal attention network (DAN; **b**, **c**), medial visual network (MVN; **d**), and salience network (SLN; **e**, **f**) at higher levels of peripheral inflammation. Scatterplots showing the strength connectivity and levels of erythrocyte sedimentation rate (ESR) are displayed below brain images. All associations were detected using seed networks identified by independent component analysis in a whole-brain search with ESR as the primary predictor of interest controlling for age and sex with $p < 0.05$ false discovery rate (FDR) cluster corrected for multiple comparisons

−0.28; both $p < 0.05$; Pearson correlation). All of the tested associations are displayed in Table 5.

## Discussion

These are some of the first analyses to identify brain regions that play an outsized role in inflammation-linked functional brain connectivity among patients with a chronic inflammatory disease. These regions were identified through converging analytic strategies. Our primary analysis used a seed network to whole-brain approach and found that multiple established intrinsic connectivity networks showed greater positive connections to the L IPL and bilateral mPFC in individuals with higher levels of inflammation. These findings were expanded upon and supported by unsupervised graph-theory-based analyses examining all pairwise connections across the whole brain. These analyses

revealed a configuration of nodes and edges that have more positive connections in individuals with higher levels of inflammation. Together these analyses suggested that some structures of the brain—particularly the L IPL—are important for the altered pattern of connectivity we observed in RA patients with higher levels of peripheral inflammation. Seed network to whole-brain analyses were replicated at the trend level 6 months later for the L IPL to DMN and for the L IPL and R mPFC to DAN, suggesting some stability of these results. Graph theoretical analyses showed that, while the L IPL and medial frontal are indeed important for inflammation-associated connectivity, they are part of a larger configuration of more positive connections involving many brain regions that participate in sustained attention and working memory. While nearly one fifth of the nodes in the whole brain participated in this configuration, only a very small fraction of possible connections were involved, and these generally involved

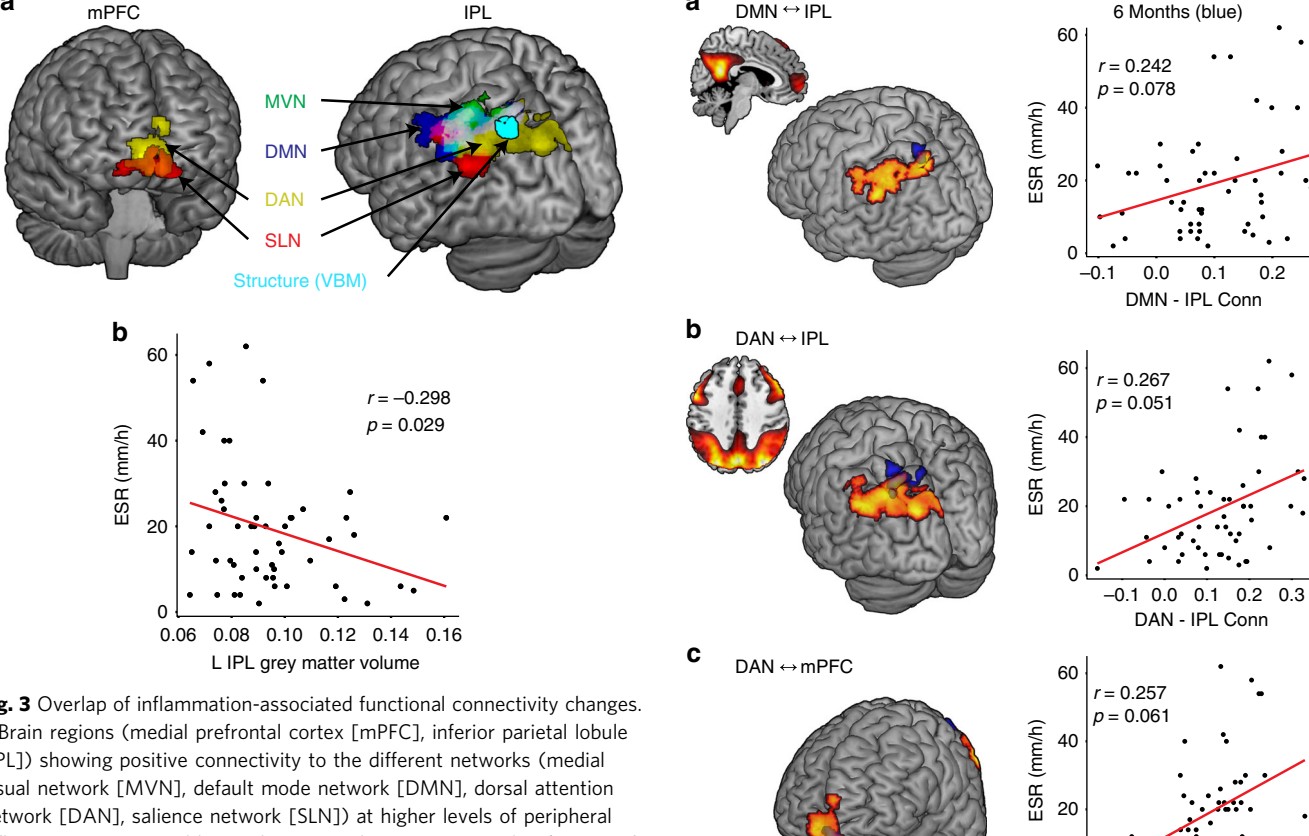

**Fig. 3** Overlap of inflammation-associated functional connectivity changes. **a** Brain regions (medial prefrontal cortex [mPFC], inferior parietal lobule [IPL]) showing positive connectivity to the different networks (medial visual network [MVN], default mode network [DMN], dorsal attention network [DAN], salience network [SLN]) at higher levels of peripheral inflammation measured by erythrocyte sedimentation rate (ESR). Area of reduced grey matter volume is shown in light blue. This was identified by testing the association of ESR with grey matter volume derived through voxel-based morphometry (VBM) in 8 mm spheres extracted around the significant peak cluster coordinates from the session 1 network to whole-brain connectivity analyses, in multiple linear regression analyses controlling for age and sex at $p < 0.05$ uncorrected. **b** Scatterplot showing the association between ESR and grey matter volume. Displayed at $p = 0.005$ voxel threshold corrected for age and sex

**Fig. 4** Overlap of inflammation-associated functional connectivity at both time points. **a** Positive connections between the default mode network (DMN) with the inferior parietal lobule (IPL), **b** dorsal attention network (DAN) and IPL, **c** the DAN and medial prefrontal cortex (mPFC) at baseline and 6 months that were associated with increased peripheral inflammation measured by erythrocyte sedimentation rate (ESR). Scatterplots showing the association between ESR and the strength of connectivity are displayed next to each brain region. Extracted average Fisher r values from 8 mm spheres derived from the significant peak cluster coordinates identified at session 1 (baseline) were used at session 2 (month 6) and these were correlated with concurrent ESR values using Pearson correlations at $p < 0.05$ uncorrected. Baseline regions are shown in orange, 6 months in blue. Displayed at $p = 0.005$ voxel threshold

more positive connections between the DMN, the DAN and the fronto-parietal task control networks. An analysis of the centrality of these nodes found that the brain regions making up the 'inflammation configuration' become increasingly prominent in the landscape of global functional connectivity in individuals with higher inflammation. Secondary analyses showed that there was also reduced GM volume in the L IPL. Finally, these patterns of inflammation-linked positive connectivity between the L IPL, bilateral mPFC and seed networks were associated with clinical symptoms of RA, including more widespread pain, fatigue, pain severity and a worse performance on the PASAT. These findings are strengthened by the agreement of some of the functional and structural outcomes and the broad agreement of analytic strategies. We conclude that the L IPL and mPFC are regions critical to the central response to peripheral inflammation in RA and may be part of a broader pattern of inflammation-linked network reorganization.

Increases in ESR and CRP are indices of overall disease activity in RA and, together with pain/tenderness and swollen joints, represent an important metric for disease severity and a benchmark for the success of treatment[16]. During heightened disease activity in RA, the balance of inflammatory factors skews towards production of pro-inflammatory mediators as a result of increased autoimmune activity[6,17]. These molecular signals subsequently reach the CNS through distal afferent nerves with cytokine receptors, sentinel immune cells embedded in the circumventricular organs outside the BBB, cytokine receptors on perivascular immune cells in the CNS and by active transport across the BBB when a sufficient circulating concentration of cytokines is reached[1]. Elevations in ESR and CRP then almost certainly represent increased activity along the various pathways[1] that induce 'sickness behaviours' under conditions of acute inflammation. Accordingly, anti-cytokine therapies represent some of the most effective and widely used treatments for RA and appear to alter CNS responses to painful stimuli[18]. In the current study, symptoms analogous to 'sickness behaviours' were associated with inflammation-linked altered connectivity. These included greater fatigue and pain intensity, more widespread pain or tenderness and a worse performance on the PASAT—positive connections between the mPFC and DAN specifically were associated with both fatigue and pain outcomes. In RA, fatigue

**Table 2 Region-of-interest functional connectivity replication analysis at 6 months**

| Seed network region (direction of association) | Pearson correlation coefficient (ESR) | p-value | Spearman's rank correlation (CRP) | p-value |
|---|---|---|---|---|
| Default mode network | | | | |
| L IPL (+) | 0.24 | 0.08 | 0.30 | 0.03 |
| R mid/sup frontal gyrus (+) | 0.23 | 0.10 | 0.22 | 0.10 |
| R posterior cingulate (−) | −0.14 | 0.32 | −0.29 | 0.04 |
| Dorsal attention network | | | | |
| L IPL (+) | 0.27 | 0.051 | 0.43 | <0.01 |
| R mPFC (+) | 0.26 | 0.061 | 0.30 | 0.03 |
| Medial visual network | | | | |
| L IPL (+) | 0.19 | 0.16 | 0.38 | <0.01 |
| Salience network | | | | |
| L mPFC (+) | 0.08 | 0.55 | 0.05 | 0.74 |
| L IPL/angular gyrus (+) | −0.05 | 0.69 | −0.05 | 0.70 |

Relationships between seed network to region connectivity and erythrocyte sedimentation rate (ESR) and C-reactive protein (CRP) at 6 months

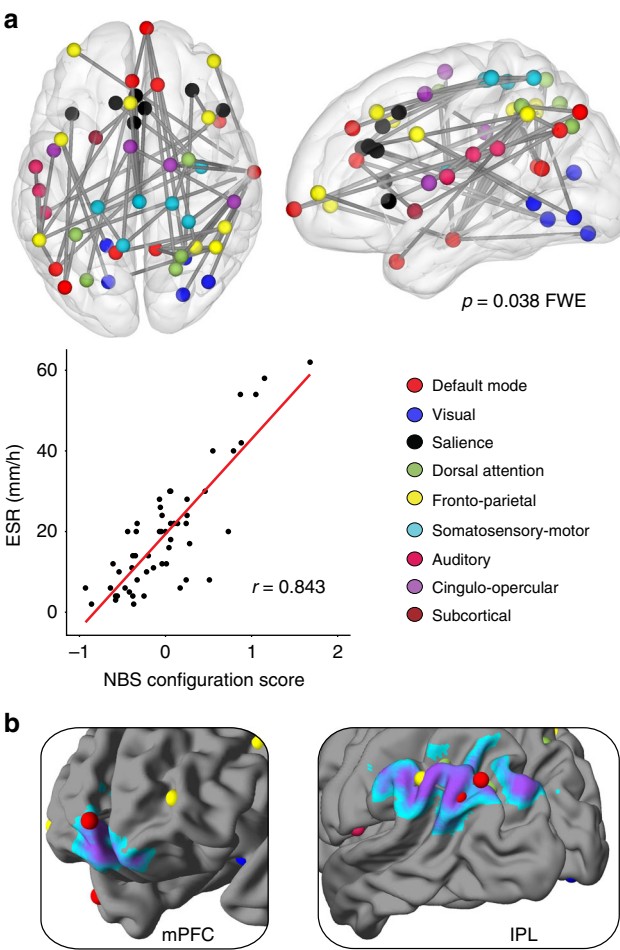

**Fig. 5** Summary of inflammation-associated functional connectivity changes. **a** Nodes and edges showing the Network-Based Statistic (NBS) inflammation configuration, with scatterplot showing the association (Pearson correlation) between erythrocyte sedimentation rate (ESR) and the average strength of connectivity across the 54 edges. Undirected and weighted Fisher z-transformed bivariate correlation matrices of connectivity were created for each subject from 10 mm diameter spheres at the 264 nodes of the Power atlas (264 × 264 matrices). Identified nodes and edges associated in aggregate with higher ESR are derived from permutation testing at family wise error corrected $p < 0.05$. **b** Overlap of regions of positive connectivity identified through seed network to whole-brain analyses and nodes identified through NBS analyses of higher inflammation

and pain are closely linked[19], so identifying common neural substrates of these symptoms could help determine whether and how anti-inflammatory treatment alters the CNS.

To date, the most compelling evidence of inflammation-linked changes in neuroimaging outcomes come from studies of healthy people given immunogenic substances such typhoid vaccine or lipopolysaccharide (LPS). Both of these substances yield transient (i.e. hours) increases in circulating levels of pro-inflammatory cytokines, such as interleukin (IL)-6[20,21] The mPFC region shown to play a role in the current study has been previously identified in studies employing experimental inflammatory paradigms. Healthy individuals administered LPS showed increased activation of the right medial frontal gyrus during social tasks with emotional valence[22,23] These findings may be relevant to the greatly impaired mental health and social function seen in chronic inflammatory conditions like RA[24]. However, in acute inflammatory paradigms with healthy individuals, distinct patterns of reduced connectivity between insula, amygdala, frontal and parietal regions have also been demonstrated using seed-based analyses[25], in contrast to the current study, where more positive connections were identified in association with higher inflammation.

Conceptually, an acute inflammatory insult would be expected to impact established neural networks to promote short-term changes in behaviour—the classic repertoire of sickness behaviours. In RA, the transient surges in inflammation occur in the context of increased basal levels of inflammation, as well as many previous inflammatory events, especially if the disease has been poorly controlled. This means the neural pathways that transduce inflammation into behaviour and symptoms in chronic conditions such as RA likely involve a more fundamental pattern of reorganization. This distinction is suggested by studies that follow the administration of potent pro-inflammatory treatments in cancer patients: fatigue and lethargy tend to develop rapidly (i.e. within 2 weeks) while more obvious neurocognitive complaints do not appear until several weeks later[9]. In the current study, we found an 'inflammation configuration' that was not particularly rich in brain regions meeting hub status—however, in individuals with higher ESR, these regions showed greater hub-like qualities, perhaps pointing to the hypothesized reorganization.

Cognitive processes supporting working memory, sustained attention and persistence through distraction involve complex interactions of neural networks. While the DMN is sometimes called a 'task-negative' network, it is becoming clear that DMN resources are recruited during complex tasks[26]. It was recently shown that, as the complexity of a working memory task increases, the global modularity of the brain decreases, meaning that the brain's subnetworks are increasingly likely to change their configurations in response to the demands of the

**Table 3 Network-Based Statistic nodes for the inflammation configuration**

| ROI | X | Y | Z | Approximate structure | Power atlas network assignment | Number of edges |
|---|---|---|---|---|---|---|
| 13[a] | −7 | −52 | 61 | L precuneus | Uncertain | 3 |
| 25[a] | 29 | −39 | 59 | R postcentral | Sensory/SM hand | 1 |
| 28[a] | 20 | −29 | 60 | R precentral | Sensory/SM hand | 2 |
| 34[a] | −21 | −31 | 61 | L postcentral | Sensory/SM hand | 3 |
| 39 | 2 | −28 | 60 | R paracentral lobule | Sensory/SM hand | 6 |
| 43 | 36 | −9 | 14 | R posterior insula | Sensory/SM mouth | 3 |
| 47[a] | −3 | 2 | 53 | L supplementary motor area | Sensory/SM mouth | 4 |
| 48 | 54 | −28 | 34 | R IPL/supramarginal gyrus | Cingulo-opercular | 1 |
| 49 | 19 | −8 | 64 | R superior frontal gyrus | Cingulo-opercular | 1 |
| 55 | −45 | 0 | 9 | L precentral/operculum | Cingulo-opercular | 1 |
| 68 | −50 | −34 | 26 | L IPL | Auditory | 2 |
| 69 | −53 | −22 | 23 | L postcentral/supramarginal gyrus | Auditory | 1 |
| 70 | −55 | −9 | 12 | L precentral/operculum | Auditory | 1 |
| 75[b] | 6 | 67 | −4 | R medial orbital frontal | Default mode | 3 |
| 82 | 46 | 16 | −30 | R superior temporal gyrus | Default mode | 1 |
| 86[c] | −44 | −65 | 35 | L angular gyrus | Uncertain | 3 |
| 87[c] | −39 | −75 | 44 | L angular gyrus | Default mode | 2 |
| 90 | −11 | −56 | 16 | L posterior cingulate/precuneus | Default mode | 1 |
| 95 | 11 | −54 | 17 | R posterior cingulate/precuneus | Default mode | 3 |
| 112[a] | −2 | 38 | 36 | L superior medial frontal gyrus | Default mode | 1 |
| 116 | 65 | −12 | −19 | R inferior temporal gyrus | Default mode | 13 |
| 122 | 12 | 36 | 20 | R dorsal anterior cingulate cortex | Default mode | 2 |
| 144 | 40 | −72 | 14 | R middle temporal gyrus | Visual | 1 |
| 150 | 27 | −59 | −9 | R parahippocampal/lingual gyrus | Visual | 2 |
| 151 | −15 | −72 | −8 | L lingual gyrus | Visual | 2 |
| 160 | −16 | −52 | −1 | L parahippocampal/lingual gyrus | Visual | 1 |
| 165 | 26 | −79 | −16 | R fusiform/lingual gyrus | Visual | 1 |
| 177[c] | −53 | −49 | 43 | L IPL | Fronto-parietal task control | 7 |
| 187 | −41 | 6 | 33 | L inferior frontal gyrus | Fronto-parietal task control | 2 |
| 190[a] | 49 | −42 | 45 | R IPL | Fronto-parietal task control | 3 |
| 192[a] | 44 | −53 | 47 | R IPL | Fronto-parietal task control | 3 |
| 196 | 40 | 18 | 40 | R middle frontal gyrus | Fronto-parietal task control | 1 |
| 197 | −34 | 55 | 4 | L middle frontal gyrus | Fronto-parietal task control | 1 |
| 199[a] | 33 | −53 | 44 | R IPL | Fronto-parietal task control | 2 |
| 200 | 43 | 49 | −2 | R orbital frontal gyrus | Fronto-parietal task control | 1 |
| 202[a] | −3 | 26 | 44 | L superior medial frontal gyrus | Fronto-parietal task control | 1 |
| 206 | 31 | 33 | 26 | R middle frontal gyrus | Salience | 2 |
| 207 | 48 | 22 | 10 | R inferior frontal gyrus | Salience | 1 |
| 208 | −35 | 20 | 0 | L anterior insula | Salience | 1 |
| 212 | −11 | 26 | 25 | L dorsal anterior cingulate cortex | Salience | 1 |
| 213[a] | −1 | 15 | 44 | L mid cingulate cortex | Salience | 1 |
| 215[a] | 0 | 30 | 27 | dorsal anterior cingulate cortex | Salience | 3 |
| 216[a] | 5 | 23 | 37 | R mid cingulate cortex | Salience | 1 |
| 227 | −22 | 7 | −5 | L putamen | Subcortical | 2 |
| 256[a] | 22 | −65 | 48 | R precuneus | Dorsal attention | 2 |
| 258[a] | 25 | −58 | 60 | R superior parietal lobule | Dorsal attention | 2 |
| 259 | −33 | −46 | 47 | L IPL | Dorsal attention | 2 |
| 260[a] | −27 | −71 | 37 | L precuneus | Dorsal attention | 1 |
| 264[a] | 29 | −5 | 54 | R middle frontal gyrus | Dorsal attention | 3 |

Nodes are identified by their parcellation according to the Power Atlas
[a]Identified as a hub by eigenvector centrality (one standard deviation above the mean)
[b]Overlap with the medial frontal region identified in seed network to whole-brain analyses
[c]Overlap with the L IPL/angular regions identified in seed network to whole-brain analyses

task[27]. Furthermore, the DMN plays a major role in this network flexibility, and the nodes that make up the DMN, along with the DAN, and fronto-parietal task control network are some of the most likely to change patterns of connectivity in response to task demands[27]. These are the same three subnetworks that comprise a significant portion of the configuration we find in the current analyses that become more positively connected to one another at higher levels of peripheral inflammation. It is possible then that the inflammation-linked changes in connectivity observed here serve a compensatory function—conscription of neural resources to perform normal cognitive tasks. This would seem to echo the overactivation seen in the aging brain during cognitive tasks[28].

Though speculative, this network conscription mechanism would be consistent with an adaptive response to impairment that does not resolve within days or hours, as it does in studies of acute inflammation.

Two findings in the current study—decreased GM volume in patients with higher peripheral inflammation and worse performance on the PASAT in association with inflammation-linked altered connectivity—are relevant in light of the mounting evidence that RA may confer an increased risk of developing dementia[29]. It is hypothesized that the basis of this risk is uncontrolled inflammatory activity, as a nested case–control study of RA patients has shown that the increased risk of

**Table 4 Region-of-interest-based analyses of grey matter volume using voxel-based morphometry**

| Region of interest | MNI coordinates (x, y, z) | B | SE | β | t | p-value |
|---|---|---|---|---|---|---|
| Default mode network | | | | | | |
| L IPL | −36, −76, 54 | −0.00032 | 0.00015 | −0.217 | −2.13 | 0.038 |
| R mid/sup frontal gyrus | 38, 52, −2 | 0.00045 | 0.00043 | 0.087 | 1.04 | 0.30 |
| R posterior cingulate | 16, −52, 30 | −0.00050 | 0.00021 | −0.240 | −2.33 | 0.024 |
| Dorsal attention network | | | | | | |
| R mPFC | 12, 62, −12 | −0.00003 | 0.00036 | −0.006 | −0.08 | 0.94 |
| Medial visual network | | | | | | |
| L IPL | −48, −56, 56 | −0.00038 | 0.00025 | −0.181 | −1.52 | 0.13 |
| Salience network | | | | | | |
| L mPFC | −2, 54, −16 | −0.00012 | 0.00035 | −0.029 | −0.32 | 0.75 |
| L IPL/angular gyrus | −52, −58, 24 | −0.00050 | 0.00050 | −0.104 | −1.00 | 0.32 |

Significance is derived from multiple linear regression models with erythrocyte sedimentation rate as the primary predictor of interest controlling for age, sex and total intracranial volume. Regions are derived from the significant seed network to whole-brain analyses

**Table 5 Pearson correlations between functional connections and clinical symptoms**

| Seed network Brain region | Fatigue | Pain intensity | % Correct (PASAT) | Diffuse pain |
|---|---|---|---|---|
| Default mode network | | | | |
| L IPL[a] | 0.19 | 0.26 | −0.19 | 0.28* |
| Dorsal attention network | | | | |
| L IPL | 0.04 | 0.18 | −0.08 | 0.22 |
| R mPFC[b] | 0.46** | 0.36** | −0.04 | 0.34* |
| Medial visual network | | | | |
| L IPL[a] | 0.16 | 0.21 | −0.08 | 0.21 |
| Salience | | | | |
| L mPFC[b] | 0.17 | 0.11 | −0.28* | 0.27* |
| L IPL/angular gyrus[a] | 0.20 | 0.13 | −0.09 | 0.34* |

*p < 0.05; **p < 0.01
[a]Overlapping regions
[b]Overlapping regions

Alzheimer's disease is attenuated significantly by anti-cytokine therapy[30]. The GM volume reduction noted in the cerebellum in association with higher peripheral inflammation was a robust finding—recent studies have found evidence that the cerebellum is selectively affected in Alzheimer's disease[31,32]. We also found, using ROI analyses based on our functional connectivity findings, that both the L IPL and posterior cingulate cortex showed reduced GM volume in individuals with higher inflammation. The L IPL and posterior cingulate are regions that have been identified using emerging positron emission tomographic imaging methods with the [11]C-PBR28 ligand, a putative marker of activated CNS microglia and astrocytes[33,34], as areas of increased neuroinflammatory activity in Alzheimer's disease[35]. Remarkably, the degree of neuroinflammation in these regions was also associated with reduced GM volume and worse neuropsychological test performance[35]. Because the L IPL and posterior cingulate were also areas that showed altered connectivity in connection with higher inflammation, it is possible that altered functional connectivity is a consequence of these structural deficits. It is even possible that these changes are early harbingers of more serious neurocognitive decline later in life.

ESR and CRP are metrics of inflammatory activity but do not show the same level of mechanistic specificity as pro-inflammatory cytokines, such as IL-6, tumour necrosis factor (TNF)-α and IL-1β. These and other inflammatory actors should

be measured in future studies of these phenomenon. The study visits did not occur within a standardized window and diurnal variation in inflammatory and imaging outcomes are possible. These findings are based on a relatively limited assessment of clinical symptoms of RA and neurocognitive performance. It is perhaps unsurprising that the subnetworks not identified in the current study are generally associated with tasks dissimilar to the PASAT, such as fine motor skills and language processing. It will be important to expand assessment in future studies to more diverse measures of neuropsychological performance and to clinical symptoms of RA, as these will likely involve their own distinct patterns of functional connectivity. Future studies in RA and other chronic inflammatory conditions should explore sex-specific differences in the impact of inflammation on neural networks as there is evidence for these differences in acute inflammatory paradigms[23]. Similarly, we used only task-based connectivity in the current analyses. A robust assessment of intrinsic/resting connectivity may produce interesting contrasts to the current findings. The NBS technique is meant to test the overall significance of a group of connections; each individual edge and node should be interpreted with caution.

These results demonstrate altered patterns of brain connectivity in RA patients with higher peripheral inflammation and may show evidence of distinct inflammation-associated subnetwork reorganization. Treatments for RA that successfully control peripheral inflammation should be investigated to determine whether they normalize brain network activity or have a neuroprotective effect. These results may have relevance for other chronic inflammatory conditions as well.

## Methods

**Design and sample.** RA patients attending a UK regional rheumatology service were consecutively approached. Those who fulfilled the 2010 American College of Rheumatology/European League Against Rheumatism classification criteria[36] and who experienced fatigue for >3 months at clinically significant levels (defined as a score of >3 on the Chalder fatigue binary scale[37]) were considered eligible for this observational cohort study. Patients with contra-indications to MRI (e.g. metal implants), alternative medical explanations for fatigue (symptomatic cardio-respiratory disease, a history of cancer in previous 5 years, unstable thyroid disease, beta-blocker prescription) or left handed were excluded. A sample size of 50 or greater was the goal of recruitment, as this provides power to detect moderate-sized correlations at relatively conservative thresholds (i.e., $r = 0.46$ for $\alpha = 0.001$), a sample size believed to perform well in the unbiased ROI-based functional connectivity analyses described below[38]. All ethical guidelines were followed during the conduct of the study and ethical approval for the study was obtained from the North of Scotland Research Ethics Committee. All participants gave full informed written consent according to the Declaration of Helsinki.

**Procedure.** All consenting participants underwent a clinical assessment phenotyping battery and multi-modal MRI brain scan. The full procedure was repeated at a second session, 6 months later.

The primary clinical outcome of interest was inflammation as measured by ESR. A measure of red blood cell aggregation, which is enhanced by inflammatory processes, ESR is one of the most widely use laboratory measures of systemic inflammation in clinical medicine and has a long standing role in aiding the diagnosis and monitoring of chronic inflammatory diseases, such as RA[39]. In RA, ESR and CRP are elevated during times of active disease, as are acute-phase pro-inflammatory cytokines, such as IL-6, TNF-α, and IL-1β[40,41]. While ESR is a relatively non-specific measure of inflammation, it does correlate moderately with almost all other metrics of inflammatory activity, including elevations of pro-inflammatory cytokines[42] and systemic glucocorticoid activation[43]. Venous blood was drawn by a trained phlebotomist as part of routine clinical practice during normal hours (approximately 9 a.m.–5 p.m.). Blood was immediately processed and analysed for the calculation of ESR (Westergren method). CRP was also analysed immediately from serum with an ADVIA® XPT immunoassay System (Siemens).

Secondary clinical outcomes of interest were (1) acute inflammation as measured by CRP, another broad marker of systemic inflammation which is routinely employed in practice. This acute phase protein is a downstream response to elevated levels of pro-inflammatory cytokines such as IL-6; it is considered a more sensitive marker of acute inflammation compared with ESR[44]. (2) Phenotypic features of sickness behaviour/clinical symptoms of RA, including (a) current levels of fatigue and pain severity, separately assessed by answering on a 0–10 numerical rating scale, (b) spatial extent of pain, as measured by the widespread pain index component of the 2011 Fibromyalgia survey criteria[45] and (c) performance (% correct) on the PASAT[46].

**MRI brain measures.** Images were acquired by a 3 Tesla, 8 channel phased array head coil using three-dimensional (3D) T1- and T2*-weighted gradient-echo echo-planar imaging (EPI) pulse sequences.

**Overview of functional connectivity.** The functional sequences were obtained using a block design where the PASAT was undertaken during 3 × 3 min 'on' periods and interspersed by 4 × 30 s rest or 'off' periods. The PASAT is a validated measure of cognitive function (specifically auditory processing, calculation, working memory and attention) which has been employed in functional MRI (fMRI) paradigms[46,47]. Participants were asked to listen to a series of numbers ranging from 1 to 9. They were required to sum consecutive numbers (i.e. the first to the second, the second to the third, etc.) and to record, via a button press, every occasion two consecutive numbers summed to the number 10. Concurrently, they were asked to focus on a computer screen displaying three boxes containing random, rapidly changing numbers. This visual stimulus was intended to distract the participants from the auditory task and hence increase difficulty. They were instructed not to process the visual numbers in any way. We chose a paradigm based on connectivity during the PASAT because this particular procedure has been shown to induce transient mental fatigue in Chronic Fatigue Syndrome[46], and in rheumatic autoimmune disease such as granulomatosis with polyangiitis[47], and thus might reflect connectivity relevant to the clinical characteristics, such as 'mental fog', in RA. It is important to note that task-based connectivity reflects both intrinsic and task-specific activity[48], and our current paradigm was not designed to dissociate these patterns.

A T2*-weighted single-shot EPI sequence [TR = 3000 ms, TE = 30 ms, flip angle (FA) = 90°, in-plane SENSE acceleration 2, matrix size 128 × 128 with 30 slices, field of view (FOV) = 240 mm, 1.88 × 1.88 × 5 mm³ voxels and 226 volumes] followed by a T1-weighted fast-field echo 3D structural scan for normalization (TR = 8.2 ms, TE = 3.8 ms, TI = 1018 ms FA = 8°, FOV = 240 mm, matrix size 240 × 240 matrix with 160 slices and 0.94 × 0.94 × 1 mm³ voxels) were acquired.

**fMRI pre-processing.** Using SPM8 (Wellcome Department of Cognitive Neurology, London, United Kingdom) running on MATLAB R2014a (Mathworks, Sherborn, MA, USA), the first four volumes were discarded to allow the magnetization to reach equilibrium, the remaining 222 functional images were realigned to the first image. The structural scans were then co-registered to a mean generated from the functional scans, after which they were segmented. The functional and structural scans were normalized to the standard SPM Montreal Neurological Institute template grey prior probability map via the individuals segmented GM image and then the functional scans were smoothed with an 8-mm full-width half-maximum (FWHM) Gaussian kernel.

**Independent component analysis.** Group ICA was performed using the Group ICA of fMRI Toolbox (GIFT) toolbox[49]. Employing pre-processed functional data from both sessions for all subjects, component estimates were validated using the Infomax ICA algorithm 20 times in ICASSO software. We used a moderate model order of 40 components to identify cortical and subcortical components that correspond to both anatomical and functional segmented networks. These included both the task and rest periods of the scan. Subject-specific spatial maps and time courses were estimated using the GICA3 back reconstruction method. Eight sub-networks identified through the ICA analyses were examined. These were the DMN, DAN, SLN, SMN, left and right fronto-parietal perception/language

networks, MVN, and the lateral visual network. These represent subnetworks identified by Beckman et al. (2005)[11] from studies of intrinsic connectivity that also strongly agree with task-based connectivity-derived networks[50]. Networks were identified and confirmed by spatial correlation between component maps and published template maps[11]. Spatial masks of the mean component map, for each intrinsic network identified, were created using the Marsbar toolbox for seed-based connectivity analyses.

**Seed network to whole-brain connectivity analysis.** Seed network to whole-brain connectivity was performed using the CONN (Cognitive and Affective Neuroscience Laboratory, Massachusetts Institute of Technology, Cambridge, MA, USA; www.nitrc.org/projects/conn) functional connectivity toolbox v15 in SPM8[51]. White matter, cerebrospinal fluid (CSF) and motion parameters were entered into the analysis as covariates of no interest. A band pass filter (frequency window: 0.01–0.1 Hz) was applied to remove linear drifts and high frequency noise from the data. The seeds used were based on the spatial masks of the intrinsic networks generated from the ICA analyses. The onsets and durations of both task 'on' and rest 'off' periods were delineated in order to measure connectivity between the networks and the rest of the brain during each condition of the scanning period. First-level analysis included bivariate correlations between voxels within each network and all voxels throughout the whole brain, thereby creating separate beta maps for each individual at each session. Only the beta maps that measured network connectivity during the PASAT task 'on' period (9 total minutes) were passed onto group second-level analyses in SPM8. The rest periods were not used in these analyses as neither the total length of 'rest' time (2 min) nor the length of individual blocks (30 s) are conducive to connectivity analyses[52].

Associations in network connectivity and ESR during the first session were assessed using a General Linear Model with age and sex entered as covariates of no interest. The resulting maps were thresholded at a whole-brain $p < 0.001$ uncorrected voxel threshold and $p < 0.05$ FDR cluster corrected for multiple comparisons. Significant regions were extracted from first-level network-to-whole-brain connectivity beta maps for each subject by obtaining the average Fisher transformed $r$ values of the identified clusters. We then performed post-hoc Spearman rank-order correlations between functional connectivity and CRP to confirm the relationships seen with ESR. Finally, replication of these significant clusters were tested with an ROI analysis by extracting the average Fisher transformed $r$ values of the 8 mm spheres—created around the significant peak cluster coordinates using the Marsbar toolbox[53]—from the relevant second session network connectivity maps and then correlating with the concurrent ESR value. These post-hoc analyses were conducted in STATA 12.1 (Stata, College Station, TX, USA), statistical significance was set at a $p$-value of 0.05 two-sided.

**Graph theoretical analyses.** Brain networks can be modelled using graph theoretical tools as a set of functional interactions made up of nodes (brain regions) and edges (functional connections between nodes)[14]. We defined the brain network using a set of 264 non-overlapping nodes based on resting-state and task functional connectivity meta-analyses[12,54]. This parcellation has been shown to produce reliable network topologies[12,55,56]. The 264 nodes were entered into the Conn Toolbox as 10 mm diameter spheres[51]. We created task-specific functional connectivity matrices by entering the onsets and durations of the PASAT task 'on' and rest 'off' periods in the Conn Toolbox as described above. A similar approach was used in a recently published study using graph theoretical analyses on task data[27]. Only the task blocks were concatenated (following our strategy in the seed network to whole-brain analyses) to create undirected and weighted Fisher $z$-transformed bivariate correlation (Pearson's $r$) matrices (264 × 264) for each participant using the average signal from each of the 264 nodes described above.

**The Network-Based Statistic.** The NBS is a statistical method for evaluating large networks, first described by Zalesky et al. (2010)[13]. In the case of functional brain connectivity, the raw data is the set of all pairwise connections (edges) between parcellated brain regions (264 regions as described above). The purpose of the NBS is to statistically control for the large number of comparisons made whenever all the individual edges of a network are tested and to isolate a configuration or configurations of nodes and edges (brain regions and connections between regions) associated with a particular research question of interest. This paradigm has been employed, for instance, in studies of schizophrenia. Statistical significance is determined through permutation testing. First, every edge (connection) is tested individually according to a pre-specified critical threshold. For the current analysis, this test was whether each and every pairwise connection (34,980) increase or decrease in strength as levels of ESR increase (controlling for age and sex). The connections that exceed this threshold are then used in a procedure analogous to cluster identification that seeks to identify a group of edges that form a graph configuration. The resulting configuration is what is statistically tested via permutation methods (5000 repetitions), and a significant family-wise error (FWE)-corrected $p$-value indicates that the configuration is associated with higher levels of ESR. Here we use a $t$-value of 3.3, corresponding approximately to a $p$-value of 0.001, as our critical threshold. FWE significance was set at $p < 0.05$.

**Hub analyses**. We calculated a measure of centrality, EC, to assess the relative global importance (hub status) of a node within the whole-brain network[15]. The EC accounts for the quantity and quality of connections by taking into account the centrality of a node and the centrality of that node's neighbours. See ref.[57] for a detailed description and mathematical formulations of this measure. EC was calculated using the Brain Connectivity Toolbox[51]. We assigned hub status to a node if the EC was greater than one standard deviation above the group mean[58]. We quantified the number of nodes in our NBS-derived configuration that met hub status by EC. To determine whether the global centrality of the nodes in the NBS configuration were associated with inflammation, we created a composite NBS EC score for each individual. This was accomplished by standardizing each EC measure for each of the 49 nodes into z-scores and averaging these for each individual. The resulting score was then correlated with ESR using Pearson correlations. The purpose of this analysis was to determine whether the nodes that make up the inflammation configuration take on a more or less prominent role in the whole-brain network topology in conjunction with rising peripheral inflammation.

**Overview of VBM**. Inspection of each T1-weighted fast-field echo data set and normalization (TR = 8.2 ms, TE = 3.8 ms, TI = 1018 ms FA = 8°, FOV = 240 mm, matrix size 240 × 240 matrix with 160 slices and $0.94 × 0.94 × 1 \, mm^3$ voxels) revealed no gross anatomical abnormalities.

**VBM pre-processing**. Inspection of each T1-weighted structural image (acquisition parameters described above) revealed no gross anatomical abnormalities. The images were then pre-processed in SPM8 (http://www.fil.ion.ucl.ac.uk/spm) using the Diffeomorphic Anatomical Registration using the Exponential Lie Algebra (DARTEL) toolbox[59]. First, the images were segmented into GM and white matter (WM) and CSF. Next, the GM probability maps were nonlinearly normalized to standard MNI space. Spatial normalization expands and contracts some brain regions; hence images were modulated in order to keep the total GM equal to the original images. The normalized and modulated images were then smoothed with an 8 mm FWHM Gaussian kernel.

**Region-of-interest VBM**. We were primarily interested in structural changes within those regions where we found a significant relationship between functional connectivity and ESR. The ROIs were defined as the 8 mm spheres created around the significant peak cluster coordinates from the session 1 network to whole-brain connectivity analyses. We extracted participant-specific raw GM values from the normalized, smoothed GM images for all of the voxels within each ROI and then performed regression analyses to examine the relationship with ESR in STATA 12.1 (adjusting for age, sex and TIV). This procedure was repeated for both session 1 and 2 data. Because we were performing a limited number of tests based on the functional connectivity findings, statistical significance was set at a p-value of 0.05 two-sided without correction.

**Whole-brain VBM**. We also conducted a whole-brain search for differences in GM volume associated with higher ESR. Smoothed GM images were entered into a multiple regression analysis in SPM8 with ESR as the covariate of interest. All analyses controlled for age, sex and TIV (the sum of GM, WM and CSF). An absolute threshold mask of 0.1 was applied to exclude edge effects and non-homogenous voxels. Results were derived from a whole-brain uncorrected voxel threshold of p < 0.001 and deemed significant at p < 0.05 FDR corrected for multiple comparisons.

**Associations with clinical symptoms**. To determine whether the significant regions identified through relationships with ESR were also related to clinical symptoms, we used significant clusters extracted from session 1 (as described above in 'seed network to whole-brain connectivity analysis') in Pearson correlation analyses with current levels of symptoms. We limited these analyses to regions showing relationships with more than one subnetwork due to the potentially large number of comparisons.

**Data availability**. The data supporting these analyses will be made available following reasonable requests to the corresponding author.

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

## Acknowledgements
Special thanks to the patient community who participated in this research effort. Thanks to Mariella D'Allesandro for efforts towards recruitment.

## Author contributions
A.S. and C.M.K. contributed equally to this work. Primary analyses and writing were performed by C.M.K., N.B., and A.S. E.I., T.L., R.E.H., S.E.H., and D.J.C. also contributed to data analysis. N.B. was responsible for primary planning and conduct of the experiment; A.D.M. and G.D.W. contributed to this effort as well. All authors contributed to the interpretation and critical revision of the manuscript.

## Additional information

**Competing interests:** Pfizer provided supplementary funding to N.B. for data acquisition. D.J.C. has consulted or served as an expert witness for Forest Laboratories, Pfizer, Inc, Cerephex Corp, Eli Lilly and Company, Merck & Co, Nuvo Research Inc, Tonix Pharmaceuticals, Johnson & Johnson, Pierre Fabre, Cypress Biosciences, Wyeth Pharmaceuticals, UCB, AstraZeneca, Jazz Pharmaceuticals, Abbott Laboratories, and Iroko Pharmaceuticals. R.E.H. has consulted for Pfizer, Inc. S.E.H. has received research funding from Aptinyx, Cerephex, Eli Lily, Forest Laboratories, and Merck and served as a consultant for Pfizer, Analgesic Solutions, Aptinyx, and deCode Genetics. A.S., C.M.K., S.E.H., R.E.H., T.L.,

E.I., and D.J.C. have all received funding from the National Institutes of Health. The remaining authors declare no competing interests.

