## [Peer Review File · Nature Communications]

Reviewers' comments:

Reviewer #1 (Remarks to the Author):

The purpose of this observational cohort study was to identify the connectivity of specific brain areas in response to peripheral inflammation in rheumatoid arthritis patients, using multi-modal MRI brain scan. This is an interesting and timely approach, given the increasing experimental and clinical evidence, that peripheral inflammatory processes contribute to the pathophysiology of neuropsychiatric and neurological disorders.

The strength of this study is certainly the reproducibility of the effects over time, reproducing the effect of the first trial six months later. However, there are also some weaknesses.

Peripheral inflammation was analyzed via erythrocyte sedimentation rate (ESR) and CRP. Both parameters are quite rough and non-specific measures of peripheral inflammation. Since empirical data indicate that neurological consequences of peripheral inflammation are mediated directly or indirectly via pro-inflammatory cytokines such as IL-6 or TNF-alpha, data of these cytokines are unfortunately lacking.

Similar to the peripheral inflammatory process, "sickness behavior" was analyzed with quite rough unspecific methods (fatigue and pain on a simple rating scale; "cognitive performance" with the PASAT). Thus, the correlational data between "neuropsychological functions" need to be interpreted with caution.

Precise information is needed regarding the medication of RA patients, since most, if not all anti-inflammatory drugs taken by RA patients induce neurological and neuropsychological unwanted side effects. Without this information, the brain scan data with the "critical inflammatory hubs" cannot be evaluated.

Information should be given regarding a possible difference between female and male patients.

Reviewer #2 (Remarks to the Author):

The authors study the relationship between chronic inflammation and brain connectivity. By studying functional connectivity to 4 seed subnetworks (DMN, DAN, SLN, SMN), they find that certain brain regions show a positive correlation between their connectivity to a subnetwork and measures of inflammation. Further, they find that, in addition to showing increased connectivity to both the DMN and DAN, the IPL also shows a negative correlation between grey matter volume and inflammation. While the basic results presented are interesting and appear to be novel, in its current form, the study appears incomplete. I have multiple concerns about the method of analysis and the conclusions drawn by the authors that need to be addressed, as I detail below.

1. Conclusion that "hubs" have been identified. Network Neuroscience is a growing area of research in which one studies the brain as a network consisting of nodes and edges. (See (Bassett & Sporns, 2017)) In this field, a network hub is a well-defined concept: a node (brain region) with exceptionally high degree compared to other nodes (van den Heuvel & Sporns, 2013). While the authors discuss brain networks and calculate functional connectivity from single seed regions, they do not perform a network analysis of the brain and do not find hub nodes in this context. I therefore find the use of the word "hub" in the title to be extremely misleading.

2. Use of subnetworks as seed regions for functional connectivity analysis. As mentioned above, performing a full network analysis to examine functional connectivity patterns in the brain is becoming a common practice – please refer to (Bullmore & Sporns, 2009) for a review. By only looking at connectivity to certain seed regions, the authors are limiting their ability to draw more meaningful conclusions about correlations between connectivity and inflammation. What if the authors had considered 8 seed subnetworks. Perhaps it might have been the case that the IPL showed increased connectivity to 6 of the subnetworks and it is the fact that there isn't an increase in connectivity to the 2 remaining subnetworks that is the interesting case. I think that the analysis and conclusions would be much more complete if the authors applied a parcellation scheme to the brain and looked at pairwise connectivity between all brain regions in the network – one can still map the network nodes to cognitive systems and study patterns of connectivity between systems (see (Muldoon et al., 2016)).

3. Choice of subnetworks used as seed regions. I do not see why it is only interesting to perform the analysis for the 4 subnetworks that the authors identify as being previously associated with inflammation. Could more cognitive systems be identified through the ICA analysis? Why not also examine those?

4. Analysis of task data. The authors cite (Cole, Bassett, Power, Braver, & Petersen, 2014) as justification for using functional connectivity during a task because it reflects both intrinsic and task based patterns of functional connectivity. However, this paper points out that while there is an underlying intrinsic network, there are specific task-based changes in functional connectivity in certain subnetworks. The authors should discuss how task performance influences the patterns of connectivity. They claim to do their analysis on both periods of task and rest data, but the results of the analysis on the rest data are never presented.

5. Analysis of structural changes. The authors limited this analysis to regions that showed correlations between connectivity and inflammation. This is a limitation of the method of analysis and use of seed regions. Using a parcellation of the brain into regions, the analysis of structural changes could be performed for all brain regions. Currently, there is no way to assess if this reduction of grey matter occurs only in this area or if it is common in other brain regions also.

6. Presentation of results.

a. In the methods section, 4 subnetworks were identified as seed regions. If no correlations were found for the SMN, this should be stated in the results and there should be discussion about what makes this subnetwork different from the others.

b. The paper contains only one figure with partial data. I see no reason to not present the data more fully. Additionally, in this figure, correlations are shown between three subnetworks and the IPL, but according to Table 2, these correlations are only present in 2 of the cases. From what I can tell, this figure is mis-labeled and not described well in the caption. Also, please use consistent ordering in Table 2 and the panels of the figure.

c. The data pertaining to the correlations between connectivity and fatigue, pain, and cognitive performance should be highlighted more – present in table form. Also, these results warrant discussion.

Minor comments:

1. Abstract, line 30 – delete “their”
2. The introduction should be expanded
3. Table 1 would be more informative if presented as a box plot figure.
4. Line 200 – should be reference to Table 2?

Bassett, D. S., & Sporns, O. (2017). Network neuroscience. *Nature Neuroscience*, 20(3), 353–364. <http://doi.org/10.1038/nn.4502>

Bullmore, E., & Sporns, O. (2009). Complex brain networks: graph theoretical analysis of structural and functional systems. *Nature Reviews Neuroscience*, 10(3), 186–198.

Cole, M. W., Bassett, D. S., Power, J. D., Braver, T. S., & Petersen, S. E. (2014). Intrinsic and Task- Evoked Network Architectures of the Human Brain. *Neuron*, 83(1), 238–251. <http://doi.org/10.1016/j.neuron.2014.05.014>

Muldoon, S. F., Pasqualetti, F., Gu, S., Cieslak, M., Grafton, S. T., Vettel, J. M., & Bassett, D. S. (2016). Stimulation-Based Control of Dynamic Brain Networks. *PLoS Computational Biology*, 12(9), e1005076. <http://doi.org/10.1371/journal.pcbi.1005076>

van den Heuvel, M. P., & Sporns, O. (2013). Network hubs in the human brain. *Trends in Cognitive Sciences*, 17(12), 683–696. <http://doi.org/10.1016/j.tics.2013.09.012>

Reviewer #3 (Remarks to the Author):

This study investigates functional and morphological neural correlates to inflammatory differences in patients with RA. The authors should be commended for choosing a both clinically and basic-science highly relevant topic, and for studying, for the circumstances, big number of patients and the aspiration to test the stability of the findings by repeating the investigation at a second time-point. The manuscript raises many questions however, which dampen my enthusiasm in spite of the interesting approach and the well-motivated study subject.

General comments

1. The manuscript lacks in transparency. This applies primarily to reporting of results, where significant findings are displayed but not non-significant ones, and where some aspects of the reporting are not clearly reported. How did the results really look like at the second scan? Were the networks reproduced etc? With an apparently large number of tests and complex research questions asked, one needs to see a fuller material to build an opinion on what is actually done and what is found. Power is neither calculated nor discussed. The choice of using task-specific activity is not fully clear to me (as role of task is not discussed and much of the comparisons seem to relate to networks that is often measured in resting-state paradigms. For example, the DMN is highly relevant in relation to a task performed; does this relate to the current results?).

2. The study question and the results need to be better put in context. Recent advances in the literature are not reflected, with relevant papers (inflammation vs brain networks, role of fatigue and pain, chronic inflammation vs glia activation etc) published e.g. in *Brain*, *Behavior*, and *Immunity* and other journals.

3. Conclusions drawn are generally much too strong, and goes from a specific method in a restricted population to broad generalizations concerning immune-to-brain relations and consequences.

Specific comments

Abstract and throughout the manuscript:

It is misleading to talk about increases, reduced grey matter, fluctuations etc, when only higher/lower levels are investigated, if not misinterpreted by me. This is between-subject differences, but can be read as relating to within-subject changes.

Introduction: RA is not included in the chronic conditions cited as relevant examples. RA is not really introduced, nor experimental research relevant for the study topic. How is ESR and CRP affected in RA?

The last sentence of the introduction can be read as relating to a hypothesis that will be tested in the study, but reflects more a framework of thinking, and needs to be rephrased.

Methods:

How many patients were not included?

How were blood samples drawn and standardized? Were diurnal variations in peripheral markers or brain correlates controlled for (or time-points at least measured)? How were analyses of peripheral markers performed?

Cognitive performance is a wide concept, but a very specific function of this is tested. Perhaps rephrase.

Rows 125-128: This information is important, but selection of these networks should be better motivated.

Results:

Does all included patients at baseline agree on and perform a second scan?
How did e.g. ESR or CRP at first and second scan correlate?

Row 188: What is Disease activity score 28?

Row 199 etc: Results from second scan need to be clearly displayed.

Discussion:

Two regions showed similar correlations to the DAN network at the second scan. Did the second show similar (but non-significant) relations as in the first scan, speaking for stability of the findings?

Row 235-238: As I know the literature of PET findings of glia activation, it is rather inconsistent. Do some earlier findings also speak against the conclusions drawn here?

We wish to thank the reviewers for their many thoughtful comments and suggestions, as well as for their overall enthusiasm for the research question. We believe that this process further strengthened the manuscript. Below we provide a brief overview of the revision before responding in detail to each comment.

We have conducted new analyses in response to the reviewers and editorial staff's requests. These allow us to address specific questions raised by the original submission, which focused upon how inflammation-associated with four seed 'regions of interest' (in this case, candidate neural networks) and their connections to the whole brain. We have now broadened our approach. Firstly, the original Independent Components Analysis (ICA) based approach has been expanded to include four additional networks. Secondly, and perhaps more consequentially, we have also conducted analyses that make use of all pairwise connections in the whole brain, or what is sometimes described as the functional 'connectome,' as suggested by the second reviewer. These analyses have identified a distinct set of brain regions (from the whole brain parcellated into 264 areas) and connections (out of the 34,000+ possibilities) associated with peripheral inflammation in this sample. We have also examined the "hub"-like qualities of the identified brain regions through graph-theory metrics. Taken together, these new results expand upon and enrich the primary analyses.

Other analyses have been conducted (including a whole-brain analysis of grey matter volume) to address concerns raised below. Finally, we have expanded the introduction and discussion sections and present the results more completely in Figures and Tables in response to several comments requesting additional content.

Please find our responses to each comment below.

Reviewer #1 (Remarks to the Author):

The purpose of this observational cohort study was to identify the connectivity of specific brain areas in response to peripheral inflammation in rheumatoid arthritis patients, using multi-modal MRI brain scan. This is an interesting and timely approach, given the increasing experimental and clinical evidence, that peripheral inflammatory processes contribute to the pathophysiology of neuropsychiatric and neurological disorders.

The strength of this study is certainly the reproducibility of the effects over time, reproducing the effect of the first trial six months later. However, there are also some weaknesses.

1. Peripheral inflammation was analyzed via erythrocyte sedimentation rate (ESR) and CRP. Both parameters are quite rough and non-specific measures of peripheral inflammation. Since empirical data indicate that neurological consequences of peripheral inflammation are mediated directly or indirectly via pro-inflammatory cytokines such as IL-6 or TNF-alpha, data of these cytokines are unfortunately lacking.

The lack of specificity of ESR and CRP do limit the peripheral mechanistic inference of the study, but there are also advantages to using these indices. They form the clinical foundation of assessing inflammatory disease activity in RA and they do show associations with the classic repertoire of pro-inflammatory cytokines that are known primarily from animal models to mediate

the neurological consequences of inflammation. We have added this information to the Methods section:

“In RA, ESR and CRP are elevated during times of active disease, as are acute-phase pro-inflammatory cytokines such as IL-6, TNF- α , and IL-1 β ,^{1,2} While ESR is a relatively non-specific measure of inflammation, it does correlate moderately with almost all other metrics of inflammatory activity, including elevations of pro-inflammatory cytokines³ and systemic glucocorticoid activation.⁴”

We have also added a brief statement to the limitations section to make clear that pro-inflammatory cytokines like IL-6 and TNF- α would be important additions to future studies.

“ESR and CRP are metrics of inflammatory activity, but do not show the same level of mechanistic specificity as pro-inflammatory cytokines such as IL-6, TNF- α , and IL-1 β . These and other inflammatory actors should be measured in future studies of these phenomenon.”

2. Similar to the peripheral inflammatory process, “sickness behavior” was analyzed with quite rough unspecific methods (fatigue and pain on a simple rating scale; “cognitive performance” with the PASAT). Thus, the correlational data between “neuropsychological functions” need to be interpreted with caution.

We agree that these symptoms represent a truncated view of both ‘sickness behaviors’ and clinical symptoms of RA, though it is encouraging to see relationships with different symptom domains (i.e. both pain and energy). We have amended the language throughout to try to focus on the specific symptoms assessed and the specific aspects of neurocognitive function measured through the PASAT. We have also added a sentence to the limitations section:

“These findings are based on a relatively limited assessment of clinical symptoms of RA and neurocognitive performance. It will be important to expand assessment in future studies to more diverse measures of neuropsychological performance and to clinical symptoms of RA, as these will likely involve their own distinct patterns of functional connectivity.”

3. Precise information is needed regarding the medication of RA patients, since most, if not all anti-inflammatory drugs taken by RA patients induce neurological and neuropsychological unwanted side effects. Without this information, the brain scan data with the “critical inflammatory hubs” cannot be evaluated.

We have added a table to the manuscript showing the breakdown of anti-inflammatory treatment for RA used by this sample. The use of medications is quite heterogeneous in this sample as we would expect. We believe the most cautious approach to these data is to simply present them to the interested reader.

The conclusion of a recent Nature Reviews Rheumatology article is that serious *neurological* side effects of biologics are quite rare (.05-.2%). *Neuropsychological impairment*, conversely, appears to be common in RA by some metrics (approximately 30%)⁵⁻⁷ though, to our knowledge, a definitive prospective study of this phenomenon has never been performed using standardized neuropsychological assessment. Moreover, it would appear that there are moderate associations between current levels of peripheral inflammation and impairment^{5,6}. Certainly the former may serve as a confounding factor in a putative link between anti-inflammatory treatments and neuropsychological impairment. An additional consideration is that

it is widely believed that most biologics do not cross the blood-brain barrier (BBB), and that even re-engineered small molecule versions of the current drugs would be unlikely to cross the BBB in pharmacologically significant amounts.⁸

4. Information should be given regarding a possible difference between female and male patients.

Because the breakdown of the current sample was 13 men and 41 women (24% versus 76%) we would be substantially underpowered to determine sex specific effects. (Please note that we adjusted for age and sex throughout the analyses).

However, as this is a very important question, we tried to determine to what degree sex may have impacted the current results by comparing the baseline unadjusted seed-network to whole brain analyses to those that we have presented in the paper (which were adjusted for sex). As you can see in the Table below, adjusting for sex appeared to have little impact on the presented results – seven of the eight the identified regions were all still associated with ESR after FDR correction when sex was not adjusted for. There was some adjustment of the peak voxel.

Seed network region (direction of association)	MNI co-ordinates (x, y, z)	Unadjusted MNI co-ordinates	p value, FDR	p value, FDR, unadjusted results
Salience Network (SLN)				
L Medial Frontal (+)	-2, 54, -16	“ “	0.009	<0.001
L IPL/ L Angular Gyrus (+)	-52, -58, 24	-40, -58, 22	0.002	0.003
Default mode network (DMN)				
L Inferior Parietal (+)	-36, -76, 54	“ ”	0.049	0.008
R Mid/Sup Frontal gyrus (+)	38, 52 -2	38, 44, 4	0.049	0.16
R Posterior Cingulate (-)	16, -52, 30	10, -46, 30	0.008	0.002
Dorsal attention network (DAN)				
L Inferior Parietal (+)	-36, -76, 54	“ “	<0.001	“ “
R Medial Frontal (+)	12, 62, -12	4, 62, -12	0.001	0.002
Medial Visual Network (MVN)				
L inferior Parietal (+)	-48, -56, 56	-48, -54, 54	0.001	0.023

We believe the most prudent thing to do is present our results as they are with adjustment for sex, while acknowledging in the limitations that sex-specific effects of inflammation should be studied further in the context of chronic inflammatory conditions.

“Future studies in RA and other chronic inflammatory conditions should explore sex-specific differences in the impact of inflammation on neural networks as there is evidence for these differences in acute inflammatory paradigms.⁹”

Reviewer #2 (Remarks to the Author):

The authors study the relationship between chronic inflammation and brain connectivity. By studying functional connectivity to 4 seed subnetworks (DMN, DAN, SLN, SMN), they find that certain brain regions show a positive correlation between their connectivity to a subnetwork and measures of inflammation. Further, they find that, in addition to showing increased connectivity to both the DMN and DAN, the IPL also shows a negative correlation between grey matter volume and inflammation. While the basic results presented are interesting and appear to be novel, in its current form, the study appears incomplete. I have multiple concerns about the method of analysis and the conclusions drawn by the authors that need to be addressed, as I detail below.

1. Conclusion that “hubs” have been identified. Network Neuroscience is a growing area of research in which one studies the brain as a network consisting of nodes and edges. (See (Bassett & Sporns, 2017)) In this field, a network hub is a well-defined concept: a node (brain region) with exceptionally high degree compared to other nodes (van den Heuvel & Sporns, 2013). While the authors discuss brain networks and calculate functional connectivity from single seed regions, they do not perform a network analysis of the brain and do not find hub nodes in this context. I therefore find the use of the word “hub” in the title to be extremely misleading.

Given the recent developments in scientific nomenclature, we fully understand why our literal use of this term may lead to confusion. For this reason we do not use the term “hub” in the title any longer. However, based on this and other helpful comments, we have conducted graph theoretical analyses in an effort to determine if the hub-like qualities of brain regions are linked to peripheral inflammation. These are discussed more fully below.

2. Use of subnetworks as seed regions for functional connectivity analysis. As mentioned above, performing a full network analysis to examine functional connectivity patterns in the brain is becoming a common practice – please refer to (Bullmore & Sporns, 2009) for a review. By only looking at connectivity to certain seed regions, the authors are limiting their ability to draw more meaningful conclusions about correlations between connectivity and inflammation. What if the authors had considered 8 seed subnetworks. Perhaps it might have been the case that the IPL showed increased connectivity to 6 of the subnetworks and it is the fact that there isn’t an increase in connectivity to the 2 remaining subnetworks that is the interesting case. I think that the analysis and conclusions would be much more complete if the authors applied a parcellation

scheme to the brain and looked at pairwise connectivity between all brain regions in the network – one can still map the network nodes to cognitive systems and study patterns of connectivity between systems (see (Muldoon et al., 2016)).

This is an excellent and thoughtful suggestion. The ICA analyses we conducted provide critical information about functional connectivity averaged over well-established neural subnetworks, and we believe they have revealed important new insights, but an unsupervised approach looking at all pairwise connectivity has great potential as well. We have expanded our analyses to include an examination of whole brain functional connectivity in a pairwise manner in the context of peripheral inflammation. These analyses were based on the parcellation scheme of Power et al. (2011).¹⁰ To address the question of how peripheral inflammation is associated with whole brain connectivity we make use of the Network Based Statistic (NBS) approach put forth by Zalesky and colleagues.¹¹ The purpose here was to identify the set of nodes (out of 264) and edges (out of 34,000 +) that are more (or less) positively connected in individuals with higher levels of ESR. We have found a set of 49 nodes and 54 edges that become more positively connected as ESR increases across the sample of RA patients. These findings also support and enrich the results of the ICA analyses. The L IPL and medial frontal regions identified in the ICA analyses were represented in the NBS derived ‘inflammation configuration,’ and the L IPL was particularly well-represented in the edges that form this pattern of altered connectivity. Additionally, the subnetworks that were associated with these regions also were well represented in the NBS. Given the large number of nodes in this parcellation scheme, and very large number of potential edges, we find it remarkable that this particular set of regions and connections were revealed – for instance, the nodes overlapping with the identified L IPL region through ICA, participate in 10 or the 54 edges identified in the NBS component (19%). We have integrated these results into the paper.

Additionally, we used graph theory metrics in relation to the NBS configuration. This analysis relates in part to the comment above about the hubs in the context of whole-brain topology. The purpose of this analysis was to determine if the global hub-like qualities of the nodes in the NBS configuration were related to ESR. We found a significant correlation between a composite measure of eigenvector centrality across all 49 nodes and ESR, such that hub-like qualities of these nodes increased in conjunction with higher ESR. This suggests that the ‘inflammation configuration,’ while itself composed of relatively few hubs, becomes more hub-like in association with higher inflammation. We now present these results in the paper as well.

3. Choice of subnetworks used as seed regions. I do not see why it is only interesting to perform the analysis for the 4 subnetworks that the authors identify as being previously associated with inflammation. Could more cognitive systems be identified through the ICA analysis? Why not also examine those?

We agree that our current knowledge of subnetworks in relation to inflammation is not complete enough to restrict our analyses to these four subnetworks. We have conducted identical analyses on an additional four networks identified through the ICA analyses. With these additional networks, our analyses now encompass the eight intrinsic connectivity subnetworks Beckman et al. (2005)¹² identified and were subsequently shown to be reproduced also by task-based connectivity.¹³ These were the medial and lateral visual networks, and the left and right frontal perception/language networks. There was indeed a significant relationship identified for the left IPL (a region overlapping with the same L IPL and L angular regions identified for the

previously reported networks) to the medial visual network. This information is now presented in full in both tables and text.

4. Analysis of task data. The authors cite (Cole, Bassett, Power, Braver, & Petersen, 2014) as justification for using functional connectivity during a task because it reflects both intrinsic and task based patterns of functional connectivity. However, this paper points out that while there is an underlying intrinsic network, there are specific task-based changes in functional connectivity in certain subnetworks. The authors should discuss how task performance influences the patterns of connectivity. They claim to do their analysis on both periods of task and rest data, but the results of the analysis on the rest data are never presented.

Our initial description of these methods was not as well articulated as it might have been. To derive the ICA networks we used all available data – both task and rest periods at both time points – but beyond this the rest periods were not analyzed. Only the task connectivity (the 9 minutes of concatenated three minute blocks of the PASAT) was used in both the ICA and graph theory analyses.

The structure of the task performed involved comparatively brief rest periods (30 seconds), four times, for a total of two minutes. Three of these “rest” periods occurred immediately after the task as well, and may well show residual activation related to the task. We believe therefore that the rest periods are likely of inadequate length to draw meaningful distinctions about task versus rest connectivity patterns and their relationship with inflammation. We have clarified our methods and have added a statement in the discussion indicating that this particular question could not be addressed in the current study.

“Similarly, we used only task-based connectivity in the current analyses. A robust assessment of intrinsic/resting connectivity may produce interesting contrasts to the current findings.”

5. Analysis of structural changes. The authors limited this analysis to regions that showed correlations between connectivity and inflammation. This is a limitation of the method of analysis and use of seed regions. Using a parcellation of the brain into regions, the analysis of structural changes could be performed for all brain regions. Currently, there is no way to assess if this reduction of grey matter occurs only in this area or if it is common in other brain regions also.

To address this concern we have performed a whole brain search for areas of decreased (and increased) grey matter as level of ESR increase across the sample. This was performed in manner analogous to the ICA analyses (i.e. controlling for age and sex and employing standards for FDR-corrected significance across the entire brain). We find one region of grey matter reduction associated with higher values of ESR using these criteria, in the left cerebellum. We report these findings in the text of the manuscript. Because this particular outcome, while interesting, does not appear to bear on the rest of the analyses in the paper, we provide only a very brief discussion of this finding. We also present all ROI/seed analyses based on the regions identified in the functional analyses in table form in the interest of completeness.

6. Presentation of results.

a. In the methods section, 4 subnetworks were identified as seed regions. If no correlations

were found for the SMN, this should be stated in the results and there should be discussion about what makes this subnetwork different from the others.

We now state clearly in text that there were 4 subnetworks that did not show associations in the current study (of eight tested), and we discuss the lack of findings for these.

b. The paper contains only one figure with partial data. I see no reason to not present the data more fully.

The paper now contains five figures in an effort to show our results more clearly. Figure 1 (as suggested) shows our clinical and symptom data as box plots. Figure 2 shows the primary results of the ICA analyses. Figure 3 shows the areas that were more positively connected to more than one network through ICA analyses. Figure 4 shows the replication analyses so that overlap across time points can be demonstrated visually. Figure 5 shows the ‘inflammation configuration’ derived from the NBS analyses, with network assignments by Power et al. (2011).

Additionally, in this figure, correlations are shown between three subnetworks and the IPL, but according to Table 2, these correlations are only present in 2 of the cases. From what I can tell, this figure is mis-labeled and not described well in the caption. Also, please use consistent ordering in Table 2 and the panels of the figure.

Our apologies for the confusion. We were inconsistent in our labeling – the left angular region from the SLN-seed analyses overlaps substantially with the L IPL regions identified in the DMN/DAN seed analyses. This was labeled as ‘L IPL’ in the Figure instead of L angular. This has been corrected. The ordering in the Table now matches that shown in the Figure.

c. The data pertaining to the correlations between connectivity and fatigue, pain, and cognitive performance should be highlighted more – present in table form. Also, these results warrant discussion.

These correlations are now presented as Table 5. We have also substantially expanded our discussion of the symptom data and its relationship to patterns of connectivity.

Minor comments:

1. Abstract, line 30 – delete “their”

This has been corrected.

2. The introduction should be expanded

The introduction has been expanded substantially.

3. Table 1 would be more informative if presented as a box plot figure.

This has been done.

4. Line 200 – should be reference to Table 2?

This should have referenced Table 2. It has been corrected.

Reviewer #3 (Remarks to the Author):

This study investigates functional and morphological neural correlates to inflammatory differences in patients with RA. The authors should be commended for choosing a both clinically and basic-science highly relevant topic, and for studying, for the circumstances, big number of patients and the aspiration to test the stability of the findings by repeating the investigation at a second time-point. The manuscript raises many questions however, which dampen my enthusiasm in spite of the interesting approach and the well-motivated study subject.

General comments

1. The manuscript lacks in transparency. This applies primarily to reporting of results, where significant findings are displayed but not non-significant ones, and where some aspects of the reporting are not clearly reported. How did the results really look like at the second scan? Were the networks reproduced etc? With an apparently large number of tests and complex research questions asked, one needs to see a fuller material to build an opinion on what is actually done and what is found.

We had initially intended our results as a concise report and therefore truncated several aspects of the manuscript. Based on these comments we have received it is clear that this work will benefit from being presented in an extended form.

Several tables have been expanded to accommodate a clear presentation of the results of the study, including non-significant findings. We now report in text the four networks (of eight tested in the current analyses) that showed no relationships on whole-brain searches with more positive values of ESR. Similarly, all of the ROI based VBM analyses are shown in table form. We also include a figure showing the overlap of the regions in the second scan, in addition to the new information derived from graph theoretical analyses.

Power is neither calculated nor discussed.

The goal of recruitment was to achieve a sample size greater than or equal to 50 participants. This decision was not based on an a priori power calculation because no study of which we are aware can provide an estimation of the effect sizes to expect for functional connectivity in the context of clinical variations in measures of peripheral inflammation. Rather, we sought to power the study for detection of moderate correlation values with seed to whole brain connectivity analyses in mind, based on Yarkoni's (2009) commentary.¹⁴ At $n=50$ the power to detect a correlation value of $r \approx .50$ is .80 at $\alpha=.001$, a reasonably conservative assumption for analysis of ROI to whole brain connectivity. As Mumford's widely cited view of power in fMRI states: 'Yarkoni (2009) suggests that sample sizes of 20 subjects used in many current fMRI studies are much too small and sample sizes of 50 or larger are most likely more appropriate to

detect the effects in fMRI.’ Our final sample size of n=54 provides power to detect a correlation of .44 at $\alpha=.001$ and .51 at $\alpha=.0001$.

We include a brief statement indicating the goal of powering the study in the Methods section:

“A sample size of 50 or greater was the goal of recruitment, as this provides power to detect moderate sized correlations at relatively conservative thresholds (i.e., $r= .46$ for $\alpha = .001$), a sample size believed to perform well in unbiased ROI-based analyses.²¹”

The choice of using task-specific activity is not fully clear to me (as role of task is not discussed and much of the comparisons seem to relate to networks that is often measured in resting-state paradigms. For example, the DMN is highly relevant in relation to a task performed; does this relate to the current results?).

This is an important point. A task was chosen because the performance of the PASAT is thought to provoke a state of mental fatigue or ‘cognitive fog,’ which represents one of the more troubling symptoms afflicting RA patients and is a core feature of inflammation induced sickness behavior. The goal was to recapitulate the patterns of connectivity that might be relevant to RA symptomology, whether that connectivity is intrinsic or evoked by the task. We have embedded an explanation of this decision in the text:

“We chose a paradigm based on connectivity during the PASAT because this particular procedure has been shown to induce transient mental fatigue in Chronic Fatigue Syndrome,¹⁵ and in other rheumatic autoimmune disease such as granulomatosis with polyangiitis,¹⁶ and thus might reflect connectivity relevant to the clinical characteristics, such as ‘mental fog,’ relevant to RA.”

Please note as well that we have clarified in response to another reviewer’s comment that task-evoked connectivity reflects a combination of intrinsic and task-based connectivity, but that we cannot distinguish between them using our current paradigm:

It is important to note that task-based connectivity reflects *both* intrinsic and task-specific activity,¹⁷ and our current paradigm was not designed to dissociate these patterns.”

We have also added a clause to the limitations section of the manuscript:

“...we used only task-based connectivity in the current analyses. A robust assessment of intrinsic/resting connectivity may produce interesting contrasts to the current findings...”

Additionally, we now provide discussion of the unique role of the DMN for demanding tasks in the Discussion section.

2. The study question and the results need to be better put in context. Recent advances in the literature are not reflected, with relevant papers (inflammation vs brain networks, role of fatigue and pain, chronic inflammation vs glia activation etc) published e.g. in Brain, Behavior, and Immunity and other journals.

We have added substantially to the discussion section with information about other studies relevant to the current analyses. These include several publications that have looked at functional imaging in the context of inflammation, and paradigms that have looked differences in symptoms in acute versus chronic inflammatory states.

3. Conclusions drawn are generally much too strong, and goes from a specific method in a restricted population to broad generalizations concerning immune-to-brain relations and consequences.

We have tried to temper our conclusions appropriately to focus on the population studied.

Specific comments

Abstract and throughout the manuscript:

It is misleading to talk about increases, reduced grey matter, fluctuations etc, when only higher/lower levels are investigated, if not misinterpreted by me. This is between-subject differences, but can be read as relating to within-subject changes.

This is an excellent point – the previous language might very easily lead to confusion. We have amended the language throughout to describe ‘higher’ rather than ‘increasing’ ESR, and ‘positive’ connections, rather than ‘increased connectivity,’ etc.

Introduction: RA is not included in the chronic conditions cited as relevant examples. RA is not really introduced, nor experimental research relevant for the study topic. How is ESR and CRP affected in RA?

We have expanded the introduction substantially, and now provide some additional context about RA as a chronic inflammatory condition. We have also made clear in the methods section how ESR and CRP relate to RA.

“In RA, ESR and CRP are elevated during times of active disease, as are acute-phase pro-inflammatory cytokines such as IL-6, TNF- α , and IL-1 β ,^{1,2} While ESR is a relatively non-specific measure of inflammation, it does correlate moderately with almost all other metrics of inflammatory activity, including elevations of pro-inflammatory cytokines³ and systemic glucocorticoid activation.⁴”

The last sentence of the introduction can be read as relating to a hypothesis that will be tested in the study, but reflects more a framework of thinking, and needs to be rephrased.

This sentence has been amended to describe the more specific goals of the current project:

“Our purpose was to identify, among patients with RA, brain regions and patterns of connectivity that are important in the central response to peripheral inflammation.”

Methods:

How many patients were not included?

A total of 335 patients were originally approached. Of these 142 were not interested, and 120 of the remaining were ineligible. Of the 73 interested and eligible participants 62 attended the first

baseline MRI session. 8 did not attend the 6 month follow up. Thus the final sample was 54. This is now shown in Supplemental Figure 1, for the interested reader. In text we report the total number first approached.

How were blood samples drawn and standardized? Were diurnal variations in peripheral markers or brain correlates controlled for (or time-points at least measured)? How were analyses of peripheral markers performed?

We have added in the requested information:

“Venous blood was drawn by a trained phlebotomist as part of routine clinical practice during normal hours (approximately 9am-5pm). Blood was immediately processed and analysed for the calculation of ESR (Westergren method). CRP was also analysed immediately from serum with an ADVIA® XPT immunoassay System (Siemens).”

Because patients were being seen in the context of clinical care, the blood draws occurred over the range of normal clinic hours. MRI sessions occurred in the same context. As diurnal variation in other indices of inflammation have been noted, and diurnal variation in patterns of brain connectivity, we have added a sentence to the limitations section:

“The study visits did not occur within a standardized window and diurnal variation in inflammatory and imaging outcomes are possible.”

Cognitive performance is a wide concept, but a very specific function of this is tested. Perhaps rephrase.

We no longer refer generically to cognitive performance but indicate that the task we used implicates sustained attention, working memory, and persistence through distraction.

Rows 125-128: This information is important, but selection of these networks should be better motivated.

At another reviewer’s request we have expanded the number of subnetworks used in the ICA analyses to include an additional four networks – these eight subnetworks represent the most stable subnetworks identified initially by Beckman et al. (2005)¹² for intrinsic connectivity, and which were subsequently shown to be stable and reproducible in the context of task-based connectivity as well.¹³ We now present these eight networks in the above context in the Methods section:

“Because the task involves sustained attention, working memory, and persistence through distraction, and is therefore described as multi-factorial,²² we examined a number of networks with cognitive performance associations. Eight subnetworks identified through the ICA analyses were examined. These were the default mode network (DMN), dorsal attention network (DAN), salience network (SLN), sensorimotor (SMN), left and right fronto-parietal networks, medial visual network, and the lateral visual network. These represent subnetworks identified by Beckman et al., (2005)²³ from studies of intrinsic connectivity that also strongly agree with task-based connectivity derived networks.²⁴”

Results:

Does all included patients at baseline agree on and perform a second scan?

Yes. As now shown in the Supplemental Figure, only those participants who completed both scans were included in these analyses.

How did e.g. ESR or CRP at first and second scan correlate?

There was relatively strong agreement between measures of ESR at time 1 and time 2 ($r = .810$, $p < .001$) and CRP at time 1 and time 2 ($\rho = .652$, $p < .001$).

Row 188: What is Disease activity score 28?

This is one of the primary measures of disease activity in rheumatoid arthritis and makes use of current peripheral inflammation (ESR or CRP levels), a count of swollen and tender joints, and a general health assessment by visual analog scale.¹⁸ We have added a brief description:

“Clinical characteristics, including ESR, CRP, Disease Activity Score -28 (a composite of swollen joint counts, tender joint counts, ESR, and self-rated health)¹⁸ and symptom levels at each time are displayed in Figure 1, and presented supplemental Table 1.”

Row 199 etc: Results from second scan need to be clearly displayed.

We now include a Figure (4) that shows the precise areas of overlap on the (trend-level) replication findings.

Discussion:

Two regions showed similar correlations to the DAN network at the second scan. Did the second show similar (but non-significant) relations as in the first scan, speaking for stability of the findings?

Yes, this is correct, and an important point for the reader.

We have added a statement to the Discussion section:

“Seed-network to whole brain analyses were replicated at the trend level six months later for the L IPL to DMN finding, and for the L IPL and medial frontal to DAN finding, suggesting some stability of these results.”

Row 235-238: As I know the literature of PET findings of glia activation, it is rather inconsistent. Do some earlier findings also speak against the conclusions drawn here?

Controversy remains not only about what pathological conditions are associated with glial activation via PET, but what the activation itself indicates. The references in the current manuscript, however, are very specifically about the ¹¹C-PBR28 ligand. It was recently shown that this specific tracer demonstrates transient increases and decreases in binding to microglia in non-human primates with paradigms validated for the depletion of microglia (colony-stimulating factor 1 receptor kinase inhibitor), and activation of microglia (LPS) respectively.¹⁹

This follows research demonstrating the specific activation of microglia (rather than astrocytes) in female baboons administered LPS.²⁰ The greatest drawback of this ligand is that it is affected by the rs6971 SNP. In both of the studies we cited looking at the tracer in the context of Alzheimer's disease^{21,22} genotype was considered in the experimental design and analyses. To our knowledge contradictory results using the ¹¹C-PBR28 ligand in dementia to examine correlates of grey matter volume and neuropsychological testing have not been published.

1. Wang, W., et al. *Rheumatology international* **32**, 887-893 (2012).
2. Brennan, F.M. & McInnes, I.B. *The Journal of clinical investigation* **118**, 3537-3545 (2008).
3. Sharma, R., et al. *Journal of the American College of Cardiology* **36**, 523-528 (2000).
4. Hardy, R., et al. *Ann Rheum Dis* **67**, 1204-1210 (2008).
5. Shin, S.Y., Katz, P., Wallhagen, M. & Julian, L. *Arthritis care & research* **64**, 1144-1150 (2012).
6. Bartolini, M., et al. *Clinical and experimental rheumatology* **20**, 491-497 (2002).
7. Appenzeller, S., Bertolo, M.B. & Costallat, L.T. *Methods and findings in experimental and clinical pharmacology* **26**, 339-343 (2004).
8. Pardridge, W.M. *Bioengineered Bugs* **1**, 231-234 (2010).
9. Eisenberger, N.I., Inagaki, T.K., Rameson, L.T., Mashal, N.M. & Irwin, M.R. *NeuroImage* **47**, 881-890 (2009).
10. Power, J.D., et al. *Neuron* **72**, 665-678 (2011).
11. Zalesky, A., Fornito, A. & Bullmore, E.T. *NeuroImage* **53**, 1197-1207 (2010).
12. Beckmann, C.F., DeLuca, M., Devlin, J.T. & Smith, S.M. *Philos Trans R Soc Lond B Biol Sci* **360**, 1001-1013 (2005).
13. Smith, S.M., et al. *Proc Natl Acad Sci U S A* **106**, 13040-13045 (2009).
14. Yarkoni, T. *Perspect Psychol Sci* **4**, 294-298 (2009).
15. Cook, D.B., O'Connor, P.J., Lange, G. & Steffener, J. *NeuroImage* **36**, 108-122 (2007).
16. Basu, N., et al. *Rheumatology (Oxford)* **53**, 2080-2087 (2014).
17. Cole, M.W., Bassett, D.S., Power, J.D., Braver, T.S. & Petersen, S.E. *Neuron* **83**, 238-251 (2014).
18. Prevoo, M.L., et al. *Arthritis and rheumatism* **38**, 44-48 (1995).
19. Hillmer, A.T., et al. *EJNMMI Research* **7**, 59 (2017).
20. Hannestad, J., et al. *NeuroImage* **63**, 232-239 (2012).
21. Kreis, W.C., et al. *Brain : a journal of neurology* **136**, 2228-2238 (2013).
22. Kreis, W.C., et al. *Neurobiology of aging* **44**, 53-61 (2016).

Reviewers' comments:

Reviewer #1 (Remarks to the Author):

My issues have been satisfactorily cleared by the authors. I do not have further comments.

Reviewer #2 (Remarks to the Author):

The authors have done an extensive amount of additional work that has greatly improved the quality of the paper and now present a much clearer description of their results.

I only have a few minor comments related to the new analysis.

In section 3.3, the authors should clarify how many of the total nodes were located in the L IPL and L angular (out of the 264). Otherwise it is hard to tell if I should be impressed that 3 NBS nodes were located in the L IPL. Is that 3 out of 12 possible L IPL nodes or 3 out of 4 possible nodes? I think that the authors are implying that a large number of the potential L IPL nodes were indicated in the NBS analysis, but this is currently not clear.

There seems to be some discrepancy between the regions that show a reduction in grey matter. In the whole brain analysis, only the left cerebellum shows a reduction, but in the ROI analysis, the L IPL and posterior cingulate cortex are associated with reduced grey matter. The authors emphasize the ROI findings in the discussion, but I am confused as to why there is not an overlap in the results of the two different methods of analysis. I therefore am skeptical of the results and associated discussion. The authors need to clarify why this discrepancy exists and either tone down the discussion of the association with grey matter or explain why it is valid to consider only the ROI results.

Reviewer #3 (Remarks to the Author):

The authors have performed impressive and substantial recalculations and a thoroughly revised manuscript. The manuscript is improved, but I still have considerations regarding conclusions drawn. The abstract should contain information about the repeated measurement (and the outcome) as this represents an important value of the design. A comment on power has been introduced, which I appreciate. On the one hand, the considerable effort of the authors to conduct a clinical study of this size with repeated measurements and complicated calculations is without question impressive, while the new analytic approach (264 parcellated areas) and connections (out of 34,000+ possibilities) introduces even more possibilities for chance findings. Perhaps such possibilities should be commented on. Conclusions should, where needed, be adjusted. One example is the first line of the discussion, which is an overstatement.

We wish to thank the reviewers for a second thoughtful review of the manuscript. We address each comment below.

Reviewer #2 (Remarks to the Author):

The authors have done an extensive amount of additional work that has greatly improved the quality of the paper and now present a much clearer description of their results. I only have a few minor comments related to the new analysis.

In section 3.3, the authors should clarify how many of the total nodes were located in the L IPL and L angular (out of the 264). Otherwise it is hard to tell if I should be impressed that 3 NBS nodes were located in the L IPL. Is that 3 out of 12 possible L IPL nodes or 3 out of 4 possible nodes? I think that the authors are implying that a large number of the potential L IPL nodes were indicated in the NBS analysis, but this is currently not clear.

There were five total nodes identified as L IPL/angular found in the NBS analysis, three of which were also identified in the ICA analyses. We can clarify that there are 8 total nodes of the 264 in the Power's atlas that are located in the L IPL and L angular. Hence, 5 of 8 potential L IPL/angular nodes were indicated in the NBS analysis. We have now introduced this detail into the manuscript.

There seems to be some discrepancy between the regions that show a reduction in grey matter. In the whole brain analysis, only the left cerebellum shows a reduction, but in the ROI analysis, the L IPL and posterior cingulate cortex are associated with reduced grey matter. The authors emphasize the ROI findings in the discussion, but I am confused as to why there is not an overlap in the results of the two different methods of analysis. I therefore am skeptical of the results and associated discussion. The authors need to clarify why this discrepancy exists and either tone down the discussion of the association with grey matter or explain why it is valid to consider only the ROI results.

We had initially conceived of the VBM analyses as secondary outcomes informed directly by our primary analyses (ICA). Our subsequent hypothesis based ROI-analysis (which tested only a small number of regions) employed uncorrected p-values (.05) to determine significance. This is a standard and widely accepted approach.¹ Conversely, the whole brain analyses require correction for multiple comparisons, as they are agnostic and undirected. The difference in statistical thresholds explains the basic discrepancy between what is found using the two approaches.

We focused on the ROI VBM analyses in the Discussion section as these were, by design, related directly to our primary analyses. However, as the findings relating to cerebellar volume were robust, we agree that it would be wise to touch on their importance in the Discussion section. We have added a sentence on the topic:

“The grey matter volume reduction noted in the cerebellum in association with higher peripheral inflammation was a robust finding – recent studies have found evidence that the cerebellum is selectively affected in Alzheimer's Disease.^{55,56”}

We have also clarified in the Methods section what motivated the ROI approach as well as the

statistical threshold used and the justification of its use. We make reference again to this approach when describing the ROI findings after the cerebellum finding. See highlighted text.

Reviewer #3 (Remarks to the Author):

The authors have performed impressive and substantial recalculations and a thoroughly revised manuscript. The manuscript is improved, but I still have considerations regarding conclusions drawn. The abstract should contain information about the repeated measurement (and the outcome) as this represents an important value of the design.

We have added a line to the abstract referencing the second fMRI session and the fact that some of the results were replicated at that time point.

“We find that higher levels of inflammation are associated with more positive connections between the inferior parietal lobule (IPL), medial prefrontal cortex, and multiple brain networks, as well as reduced IPL grey matter. When the procedure is repeated six months later, some of the same patterns of connectivity are again associated with higher peripheral inflammation.”

A comment on power has been introduced, which I appreciate. On the one hand, the considerable effort of the authors to conduct a clinical study of this size with repeated measurements and complicated calculations is without question impressive, while the new analytic approach (264 parcellated areas) and connections (out of 34,000+ possibilities) introduces even more possibilities for chance findings. Perhaps such possibilities should be commented on.

We have added a line to the Limitation section acknowledging the need for caution when interpreting individual nodes and edges in the new analytic technique (the NBS):

“The NBS technique is meant to test the overall significance of a group of connections; each individual edge and node should be interpreted with caution.”

Conclusions should, where needed, be adjusted. One example is the first line of the discussion, which is an overstatement.

We have amended this line to acknowledge that our study is *one of the first* to examine this question in a chronic inflammatory disease, rather than *the first*, which may not be accurate. Other, minor adjustments of language have occurred as well.

1. Whitwell, J.L. Voxel-based morphometry: an automated technique for assessing structural changes in the brain. *J Neurosci* **29**, 9661-9664 (2009).

REVIEWERS' COMMENTS:

Reviewer #2 (Remarks to the Author):

The authors addressed my comments in their response letter, but I found that the added text in the manuscript did not explain to the reader that the discrepancy between the detected ROIs that show a reduction in grey matter was due to a difference in the stringency of the statistical testing applied. It is true that the criteria for each analysis are stated in the methods, but I believe that it is worth being very upfront with readers about differences in results due to different statistical tests. I do not assume that readers will automatically realize that, in one case, values were corrected for multiple comparisons, and in another they were not. It is my opinion that it is important to also report trends in data – statistical significance is manipulated far too often – so if the ROI analysis didn't pass significance tests with correction for multiple comparisons, the authors should state this. The findings are still important.

Reviewer #3 (Remarks to the Author):

I am happy with the revision.

[REVIEWER COMMENT]

The authors addressed my comments in their response letter, but I found that the added text in the manuscript did not explain to the reader that the discrepancy between the detected ROIs that show a reduction in grey matter was due to a difference in the stringency of the statistical testing applied. It is true that the criteria for each analysis are stated in the methods, but I believe that it is worth being very upfront with readers about differences in results due to different statistical tests. I do not assume that readers will automatically realize that, in one case, values were corrected for multiple comparisons, and in another they were not. It is my opinion that it is important to also report trends in data – statistical significance is manipulated far too often – so if the ROI analysis didn't pass significance tests with correction for multiple comparisons, the authors should state this. The findings are still important.

To address this remaining concern we have directly stated that this result did not meet correction for multiple comparisons when describing the results of both the baseline and six-month ROI structural analyses:

"This result did not meet false discovery rate (FDR) correction, a method for controlling for multiple comparisons. ($p > .05$ FDR)."

"Similar to the baseline results, this finding in the L IPL did not meet FDR correction ($p > .05$ FDR)."